# CaliDrop: KV Cache Compression with Query-based Calibration

## Abstract

Large Language Models (LLMs) require substantial computational resources during generation. While the Key-Value (KV) cache significantly accelerates this process by storing attention intermediates, its memory footprint grows linearly with sequence length, batch size, and model size, creating a bottleneck in long-context scenarios. Various KV cache compression techniques, including token eviction, quantization, and low-rank projection, have been proposed to mitigate this bottleneck, often complementing each other. This paper focuses on enhancing token eviction strategies. Token eviction leverages the observation that the attention patterns are often sparse, allowing for the removal of less critical KV entries to save memory. However, this reduction usually comes at the cost of notable accuracy degradation, particularly under high compression ratios. To address this issue, we propose **CaliDrop**, a novel strategy that enhances token eviction through calibration. Our preliminary experiments show that queries at nearby positions exhibit high similarity. Building on this observation, CaliDrop performs speculative calibration on the discarded tokens to mitigate the accuracy loss caused by token eviction. Extensive experiments demonstrate that CaliDrop significantly improves the accuracy of existing token eviction methods [1].

## 1 Introduction

Large language models (LLMs) have demonstrated remarkable capabilities across various domains (Achiam et al., 2023; Touvron et al., 2023; Dubey et al., 2024; Jiang et al., 2023; Liu et al., 2024b). However, their auto-regressive generation process often results in slow inference speeds. KV cache reduces the computational complexity of self-attention from $O(n^2)$ to $O(n)$ by storing intermediate key-value pairs from previous attention operations. This allows the model to avoid recalculating these values in future attention computations, thus speeding up inference. However, this efficiency comes at the cost of additional memory usage, as the KV cache must store the key-value pairs for each token. The memory overhead of the KV cache grows linearly with the sequence length, which becomes particularly severe in long-context scenarios (Xiao et al., 2024), posing substantial deployment challenges and creating the need for effective KV cache compression solutions.

Existing KV cache compression methods include quantization (Kang et al., 2024; Liu et al., 2024e), layer-wise sharing (Wu & Tu, 2024; Brandon et al., 2024), prefix sharing (Juravsky et al., 2024; Zhu et al., 2024), head-wise sharing (Shazeer, 2019; Ainslie et al., 2023), token eviction (Li et al., 2024; Zhang et al., 2023), and low-rank projection (Wang et al., 2024a; Yu et al., 2024). Our work focuses on token eviction, which reduces memory consumption by removing non-critical tokens from the KV cache. Current token eviction methods typically leverage the observation that a small subset of tokens contributes to the majority of attention scores. For example, StreamingLLM (Xiao et al., 2024) argues that the tokens at the beginning and those closest to the current token have higher attention scores. H2O (Zhang et al., 2023) estimates the importance of each token by using the accumulated attention score. SnapKV (Li et al., 2024) estimates token importance using an observing window and the pooled accumulated attention score. However, these approaches face two fundamental limitations in high compression ratio scenarios: (1) Permanently discarding tokens can be detrimental to accuracy if those tokens later become crucial, and (2) Although the removed tokens are not important, the accumulated effect of numerous discarded tokens cannot be overlooked (Yang et al., 2024b).

---

[1] Code is available at https://anonymous.4open.science/r/CaliDrop.

To address these limitations, we propose CaliDrop, a novel strategy that enhances token eviction methods through query-based calibration. Our key insights are based on two empirical observations: (1) Queries at nearby positions exhibit high similarity, and (2) Historical attention outputs can be used to predict future attention outputs. Building on these insights, CaliDrop compensates for evicted tokens by precomputing attention outputs for queries at nearby positions, alleviating memory pressure while maintaining model accuracy. Extensive experiments demonstrate that CaliDrop significantly enhances the accuracy of existing token eviction methods with a manageable cost in efficiency.

In summary, our contributions are as follows:

- We analyze the similarity between queries of the model and find that queries at nearby positions typically exhibit higher cosine similarity.
- We propose CaliDrop, a general strategy that can be integrated with existing token eviction methods to effectively approximate compensation for evicted tokens, thus improving accuracy under high compression ratios.
- Experiments on different benchmarks, models, and compression ratios have demonstrated that our method significantly improves accuracy over existing token eviction methods.

## 2 RELATED WORK

**KV Cache Compression.** KV cache compression is essential for reducing memory usage and improving efficiency in Large Language Models, allowing faster inference and deployment on resource-constrained scenarios. There are different methods for KV cache compression. Low-rank approximation (Kang et al., 2024; Chang et al., 2024) compresses KV cache by leveraging their low-rank structure. Group Query Attention (GQA) (Ainslie et al., 2023) and Multi-Query Attention (MQA) (Shazeer, 2019) share KV cache within different heads to reduce redundancy. Multi-Head Latent Attention (MLA) (Liu et al., 2024a) down-projects KV cache to low-rank spaces for memory optimization. Token eviction (Zhang et al., 2023; Xiao et al., 2024; Li et al., 2024) discard some unimportant tokens during generation, in order to reduce the size of the KV cache while preserving higher accuracy as much as possible. KV cache quantization (Sheng et al., 2023; Liu et al., 2024e; Hooper et al., 2024; He et al., 2024; Zhang et al., 2024a; Dong et al., 2024; Yang et al., 2024b) stores unimportant tokens in lower precision and retains important ones in full precision. Inter-layer KV sharing (Sun et al., 2024; Wu & Tu, 2024; Brandon et al., 2024) reduces memory usage by allowing different layers to share the same KV cache, significantly lowering redundancy across layers. Prefix sharing (Juravsky et al., 2024; Zhu et al., 2024) shares common prefixes among different sequences to reduce redundancy. KV merging (Wang et al., 2024b; Zhang et al., 2024c) uses reparameterization and interpolation techniques to reduce redundancy while preserving semantic integrity. These approaches are often orthogonal and can typically be combined to achieve superior compression results and efficiency gains.

**Token Eviction.** To reduce memory usage, eviction-based strategies keep a fixed KV cache size to store critical KV pairs and discard unnecessary pairs. Most methods evaluate token importance based on attention scores. As demonstrated in (Liu et al., 2024d; Zhang et al., 2023; Li et al., 2024), tokens with higher accumulated attention scores are considered more important. Alternative strategies employ factors such as initial tokens (Xiao et al., 2024), special tokens (Ge et al., 2023; Chen et al., 2024), or L2 norms (Devoto et al., 2024). Recent studies focus on optimizing the allocation of the KV cache memory budget. Some focus on cross-layer strategies, where PyramidKV (Cai et al., 2024) and PyramidInfer (Yang et al., 2024a) utilize a pyramid-shaped memory allocation while selecting tokens with high attention scores in each layer. CakeKV (Qin et al., 2025) dynamically analyzes the attention patterns of each layer during the prefill stage to adaptively allocate KV cache sizes. DynamicKV (Zhou et al., 2024) computes the average attention scores of recent and historical tokens, allocating budgets proportionally based on the density of critical tokens in each layer. Other works focus on head-level allocation strategies. AdaKV (Feng et al., 2024) optimizes L1 loss between original and pruned multi-head attention outputs for head-wise budget assignment, achieving a better budget allocation. LeanKV (Zhang et al., 2024b) optimizes KV cache management through heterogeneous quantization, dynamic sparsity, and unified compression. HeadKV (Fu et al., 2024) evaluates the retrieval and reasoning performance to optimize the KV cache allocation.

## 3 METHOD

### 3.1 PRELIMINARY EXPERIMENTS

While prior studies indicate historical attention scores can approximate future patterns (Zhang et al., 2023), the predictive power of historical queries and the utility of historical attention outputs for future approximations remain less explored. To establish a foundation for our proposed method, we conduct preliminary experiments addressing two questions: (1) Can historical queries approximate future queries? (2) Can historical attention outputs estimate future attention outputs?

#### 3.1.1 CAN HISTORICAL QUERIES APPROXIMATE FUTURE QUERIES?

To investigate whether historical queries can approximate future queries, we conduct experiments using LLaMA-3-8b-Instruct (Dubey et al., 2024) on LongBench (Bai et al., 2024). We retain all queries of Layer 10, Head 16 during inference and calculate the cosine similarity between each of them. Figure 1 (top-right) presents a heatmap of the cosine similarity of queries in a single sample (we plot the results of token positions from 50-150 for simplicity). While we show only one figure for clarity, similar trends are observed across other layers, heads, tokens and samples. The heatmap indicates that queries at nearby positions exhibit high cosine similarity, suggesting that historical queries indeed provide a reliable approximation for future queries.

#### 3.1.2 ATTENTION DECOMPOSITION THEOREM

Before answering Q2, we propose a theoretical foundation for splitting attention computation. Let $Q$ be the Query of the current tokens, and let $K$, $V$ represent the Key and Value of a set of $n$ tokens $S = \{t_1, \ldots, t_n\}$ in the KV cache. Consider a partition of the tokens into disjoint subsets $S_i$ and $S_j$ such that $S_i \cup S_j = S$. Let $K_{S_i}, V_{S_i}$ and $K_{S_j}, V_{S_j}$ be the corresponding subset of $K$ and $V$. The attention computation over the full set $S$ can be decomposed as follows:

$$\text{Att}(Q, K, V) = \alpha_i \cdot \text{Att}(Q, K_{S_i}, V_{S_i}) + \alpha_j \cdot \text{Att}(Q, K_{S_j}, V_{S_j}), \tag{1}$$

where the weights $\alpha_i$ and $\alpha_j$ represent the normalized exponential sum of the attention weighs:

$$\alpha_i = \frac{\sum_{t \in S_i} e^{\frac{Q K_t^\top}{\sqrt{d_k}}}}{\sum_{t \in S} e^{\frac{Q K_t^\top}{\sqrt{d_k}}}}, \quad \alpha_j = \frac{\sum_{t \in S_j} e^{\frac{Q K_t^\top}{\sqrt{d_k}}}}{\sum_{t \in S} e^{\frac{Q K_t^\top}{\sqrt{d_k}}}}. \tag{2}$$

Note that $\alpha_i + \alpha_j = 1$. This decomposition allows for the independent attention computation of the subsets and is crucial for our hybrid attention computation strategy.

***Proof.*** 
$$\text{Att}(Q, K, V) = \text{softmax}\left(\frac{QK^\top}{\sqrt{d_k}}\right) V = \frac{\sum_{t \in S} e^{\frac{Q K_t^\top}{\sqrt{d_k}}} V_t}{\sum_{t \in S} e^{\frac{Q K_t^\top}{\sqrt{d_k}}}} \tag{3}$$

$$= \frac{\sum_{t \in S_i} e^{\frac{Q K_t^\top}{\sqrt{d_k}}} V_t + \sum_{t \in S_j} e^{\frac{Q K_t^\top}{\sqrt{d_k}}} V_t}{\sum_{t \in S} e^{\frac{Q K_t^\top}{\sqrt{d_k}}}}$$

$$= \frac{\sum_{t \in S_i} e^{\frac{Q K_t^\top}{\sqrt{d_k}}}}{\sum_{t \in S} e^{\frac{Q K_t^\top}{\sqrt{d_k}}}} \cdot \frac{\sum_{t \in S_i} e^{\frac{Q K_t^\top}{\sqrt{d_k}}} V_t}{\sum_{t \in S_i} e^{\frac{Q K_t^\top}{\sqrt{d_k}}}} + \frac{\sum_{t \in S_j} e^{\frac{Q K_t^\top}{\sqrt{d_k}}}}{\sum_{t \in S} e^{\frac{Q K_t^\top}{\sqrt{d_k}}}} \cdot \frac{\sum_{t \in S_j} e^{\frac{Q K_t^\top}{\sqrt{d_k}}} V_t}{\sum_{t \in S_j} e^{\frac{Q K_t^\top}{\sqrt{d_k}}}}$$

$$= \alpha_i \cdot \text{Att}(Q, K_{S_i}, V_{S_i}) + \alpha_j \cdot \text{Att}(Q, K_{S_j}, V_{S_j})$$

#### 3.1.3 CAN HISTORICAL ATTENTION OUTPUTS ESTIMATE FUTURE ATTENTION OUTPUTS?

In this part, we investigate whether historical attention outputs can be used to approximate future attention outputs. Directly multiplying the historical query with the KV cache is not feasible, as

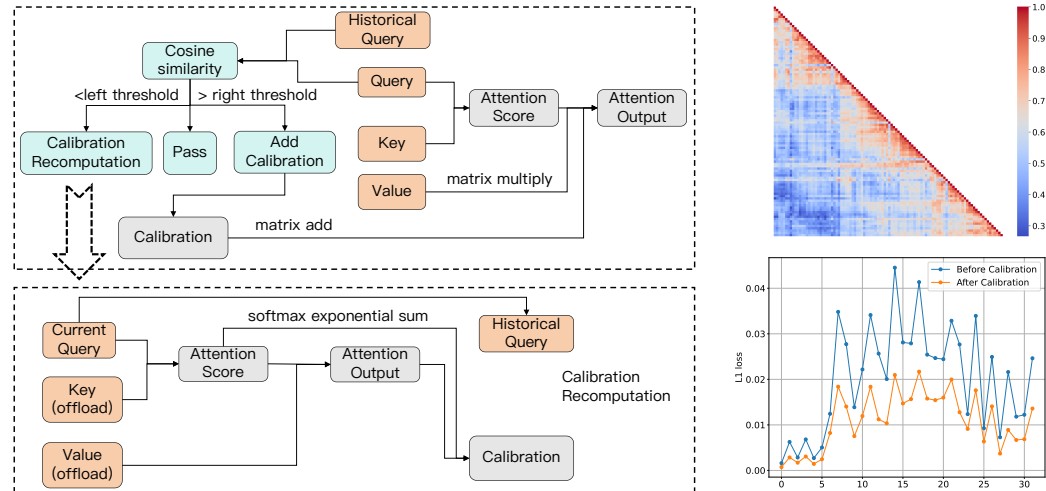

Figure 1: Illustration of the proposed method and preliminary results. Left: Overview of the method in decoding. We first compute the cosine similarity between the current query and historical query, and then decide whether to perform recomputation, apply calibration, or take no action based on the similarity. Top-right: the heatmap of cosine similarity of the queries in an example. Bottom-right: L1 loss before and after calibration. The loss significantly decreases after calibration.

it would discard the information from the current query entirely. Therefore, we propose a method that splits the KV cache into two parts (important & unimportant) inspired by the token eviction methods. We compute the current query with the important KV cache and the historical query with the unimportant KV cache. We integrate the two results according to Equation 1 as the final attention output to approximate the true current attention output. We refer to this integration process as **calibration**, which is the core of our method.

To preliminarily validate the effectiveness of this method, we conduct experiments using the LLaMA-3-8b-Instruct (Dubey et al., 2024) model on the LongBench (Bai et al., 2024) benchmark. We use the token eviction strategy from SnapKV (Li et al., 2024) to select 128 important tokens and other tokens are considered unimportant. We set the current token to be the first token after the prefill phase. We set the historical token to be one of the 0-9th tokens preceding the current token (randomly selected). We compare the L1 loss of the approximated attention output and the true current attention output before and after calibration across different layers.

Theoretically, if the historical query is completely the same as the current query, the L1 loss should be 0. When the historical and current queries are highly similar, we expect the calibration to provide a positive effect. Conversely, if the queries are not similar, there will be a side effect. Figure 1 (bottom-right) shows the L1 loss before and after calibration across different layers. The results demonstrate that adding calibration significantly reduces the L1 loss. Similar trends are observed across other tokens and samples, although additional figures are omitted for brevity. From these findings, we conclude that historical attention outputs can help approximate future attention outputs.

## 3.2 CALIDROP

After the aforementioned preliminary results, we introduce CaliDrop. CaliDrop can be applied on top of existing token eviction methods to enhance accuracy. Following SnapKV Li et al. (2024), we focus on token eviction during prefilling, although our method can be easily extended to the decoding phase. As shown in Figure 1 (left), after token eviction based on a certain strategy, we do not directly delete the evicted KV cache. Instead, we offload them. At this point, we use the current query to perform attention computation with the evicted KV cache. After computation, we retain the current query, the softmax denominator, and the attention output for future potential calibration.

In the decoding phase, we add calibration according to Equation 1. However, the similarity between the current query and the historical query decreases as the positional difference increases. Thus, we need to ensure that the added calibration mostly provides a positive effect. To address this, we

introduce two hyper-parameters: the left threshold ($\theta_1$) and the right threshold ($\theta_2$). These thresholds control whether recomputation should be performed and calibration should be applied. We calculate the cosine similarity between the current query and the retained historical query. If the cosine similarity is lower than $\theta_1$, we consider the current query and historical query to be sufficiently distant. In this case, we reload the evicted KV cache, recompute the attention output with the current query, and update the historical query, the softmax denominator, and the attention output. This hyper-parameter controls the frequency of recomputation, thus controlling the additional computation overhead. If the cosine similarity is higher than $\theta_2$, we assume the current query and the historical query are sufficiently similar, and we add calibration because we believe this calibration is likely to improve accuracy. When the cosine similarity lies between the two thresholds, no recomputation or calibration is performed. $\theta_1$ and $\theta_2$ work together to balance the accuracy of calibration and the additional computational overhead. The process of CaliDrop are formally described in Algorithms 1

---

**Algorithm 1** The CaliDrop Algorithm

---

**Require:** Input sequence $X$, eviction strategy $f$, thresholds $\theta_1 < \theta_2$

1: **– Prefilling Phase –**

2: Compute $Q, K, V$ and partition into $KV_{\text{compress}}, KV_{\text{evict}}$ via strategy $f$.

3: Using the last query $Q_{-1}$, cache initial calibration values:

4:     $C_{weight} \leftarrow \sum \exp\left(\frac{Q_{-1} K_{\text{evict}}^{\top}}{\sqrt{d_k}}\right)$               $\triangleright$ Exponential sum of attention weights

5:     $C_{out} \leftarrow \text{Att}(Q_{-1}, K_{\text{evict}}, V_{\text{evict}})$             $\triangleright$ Calibration output

6:     $C_q \leftarrow Q_{-1}$                        $\triangleright$ Calibration query

7: **– Decoding Phase –**

8: **for** each new token to be generated $t = 1, 2, \ldots$ **do**

9:     $\text{Output}_t, A_{weight} \leftarrow \text{Att}(Q_t, K_{\text{compress}}, V_{\text{compress}})$

10:     $\rho \leftarrow \cos(Q_t, C_q)$           $\triangleright$ Compute the cosine similarity of the queries

11:     **if** $\rho < \theta_1$ **then**

12:         $C_{weight} \leftarrow \sum \exp\left(\frac{Q_t K_{\text{evict}}^{\top}}{\sqrt{d_k}}\right)$         $\triangleright$ Recompute calibration with $Q_t$

13:         $C_{out} \leftarrow \text{Att}(Q_t, K_{\text{evict}}, V_{\text{evict}})$

14:         Update historical query: $C_q \leftarrow Q_t$

15:         $\text{Output}_t \leftarrow \text{Calibrate}(\text{Output}_t)$           $\triangleright$ Apply calibration

16:     **else if** $\rho > \theta_2$ **then**

17:         $\text{Output}_t \leftarrow \text{Calibrate}(\text{Output}_t)$           $\triangleright$ Apply calibration

18:     **end if**

19: **end for**

---

## 4 EXPERIMENTS

### 4.1 EXPERIMENT SETUP

To verify the effectiveness of our method, we conduct extensive experiments on a series of benchmarks and models. The detailed experimental settings are as follows.

#### 4.1.1 MODELS AND BENCHMARKS

We test our method on three LLMs: Mistral-7B-Instruct (Jiang et al., 2023), LLaMA-3-8B-Instruct, and LLaMA-3-70B-Instruct (Dubey et al., 2024). These models are chosen for their strong performance and wide utilization in natural language processing tasks.

To thoroughly evaluate the capabilities of these models with different methods, we use three benchmarks that focus on different aspects of language understanding and reasoning.

Table 1: Performance comparison of CaliDrop with SnapKV, H2O, StreamingLLM (SLM) and FullKV on LongBench for LLaMA-3-8B-Instruct and Mistral-7B-Instruct. CaliDrop generally achieves improvements over previous KV cache compression methods across various KV cache sizes and LLMs. The performance strengths of CaliDrop are more evident in small KV cache sizes (i.e. KV Size = 64). **Bold** represents the best performance.

| Method | Single-Document QA | | Multi-Document QA | | Summarization | | Few-shot Learning | | | Synthetic | | Code | | Avg. |
|---|---|---|---|---|---|---|---|---|---|---|---|---|---|---|
| | MF-en | Qasper | HotpotQA | 2WikiMQA | GovReport | MultiNews | TREC | TriviaQA | SAMSum | PCount | PRe | Lcc | RB-P | |
| | 18409 | 3619 | 9151 | 4887 | 8734 | 2113 | 5177 | 8209 | 6258 | 11141 | 9289 | 1235 | 4206 | |
| *Mistral-7B-Instruct, KV Size = Full* | | | | | | | | | | | | | | |
| FullKV | 48.54 | 24.35 | 32.92 | 21.87 | 33.05 | 25.77 | 67.00 | 86.84 | 40.95 | 5.40 | 91.00 | 57.24 | 49.84 | 44.98 |
| *Mistral-7B-Instruct, KV Size = 64* | | | | | | | | | | | | | | |
| SLM | 22.76 | 10.31 | 16.59 | 14.09 | 13.41 | 12.21 | 35.67 | 75.93 | 26.47 | 3.66 | 64.60 | 41.27 | 35.41 | 28.64 |
| +CaliDrop | 29.62 | 13.29 | 22.54 | 15.70 | 20.09 | 19.14 | 54.00 | 78.68 | 30.71 | 7.09 | 63.56 | 44.62 | 37.99 | **33.62** |
| H2O | 32.91 | 13.15 | 17.60 | 14.29 | 19.56 | 18.99 | 39.33 | 79.77 | 35.80 | 5.09 | 74.76 | 48.19 | 40.14 | 33.81 |
| +CaliDrop | 37.39 | 17.73 | 22.19 | 16.93 | 23.03 | 22.01 | 56.00 | 83.69 | 37.12 | 5.32 | 77.89 | 49.31 | 42.88 | **37.81** |
| SnapKV | 34.46 | 12.87 | 20.11 | 15.52 | 17.33 | 16.37 | 39.00 | 81.14 | 35.14 | 6.89 | 73.55 | 47.05 | 39.80 | 33.79 |
| +CaliDrop | 39.16 | 16.94 | 22.90 | 17.67 | 21.83 | 21.38 | 55.33 | 84.25 | 37.09 | 7.17 | 80.33 | 46.72 | 41.93 | **37.90** |
| *Mistral-7B-Instruct, KV Size = 128* | | | | | | | | | | | | | | |
| SLM | 22.66 | 8.23 | 18.15 | 14.98 | 14.72 | 14.25 | 41.00 | 79.27 | 34.82 | 3.74 | 40.32 | 48.47 | 39.08 | 29.21 |
| +CaliDrop | 30.34 | 12.31 | 22.55 | 16.61 | 20.47 | 20.39 | 53.33 | 79.95 | 36.03 | 7.14 | 54.29 | 50.62 | 41.58 | **34.28** |
| H2O | 32.94 | 13.75 | 18.72 | 15.08 | 21.20 | 20.71 | 43.00 | 82.20 | 37.69 | 6.38 | 82.50 | 51.3 | 42.55 | 36.00 |
| +CaliDrop | 37.74 | 18.60 | 23.47 | 17.62 | 23.78 | 22.37 | 55.67 | 85.54 | 38.88 | 5.95 | 83.92 | 51.50 | 44.86 | **39.22** |
| SnapKV | 37.44 | 14.85 | 21.45 | 15.48 | 20.76 | 20.15 | 47.33 | 84.46 | 37.34 | 5.56 | 85.22 | 51.04 | 42.50 | 37.20 |
| +CaliDrop | 41.83 | 18.75 | 24.52 | 18.12 | 23.68 | 22.23 | 56.33 | 86.19 | 37.90 | 5.63 | 86.14 | 51.68 | 44.53 | **39.81** |
| *LLaMA-3-8B-Instruct, KV Size = Full* | | | | | | | | | | | | | | |
| FullKV | 40.56 | 37.54 | 49.81 | 34.93 | 31.04 | 25.58 | 69.67 | 89.85 | 40.50 | 12.94 | 83.67 | 56.58 | 51.01 | 47.98 |
| *LLaMA-3-8B-Instruct, KV Size = 64* | | | | | | | | | | | | | | |
| SLM | 24.73 | 20.83 | 43.86 | 29.93 | 14.59 | 13.50 | 36.67 | 71.49 | 25.80 | 14.67 | 81.00 | 49.67 | 42.81 | 36.12 |
| +CaliDrop | 31.52 | 27.21 | 46.79 | 32.17 | 17.27 | 17.27 | 60.00 | 81.60 | 29.45 | 12.33 | 78.00 | 49.39 | 43.79 | **40.52** |
| H2O | 31.14 | 24.28 | 47.41 | 31.25 | 19.02 | 18.95 | 38.33 | 86.77 | 35.58 | 8.69 | 82.33 | 57.38 | 48.44 | 40.74 |
| +CaliDrop | 35.92 | 28.80 | 49.58 | 32.83 | 20.85 | 20.86 | 56.00 | 88.08 | 36.95 | 11.44 | 82.67 | 55.75 | 47.92 | **44.13** |
| SnapKV | 33.33 | 25.05 | 47.56 | 31.83 | 16.79 | 17.20 | 40.67 | 86.06 | 33.81 | 12.22 | 77.67 | 56.27 | 48.07 | 40.50 |
| +CaliDrop | 36.91 | 31.27 | 49.33 | 33.97 | 19.16 | 19.73 | 62.33 | 88.10 | 35.56 | 11.55 | 78.67 | 54.85 | 45.55 | **43.61** |
| *LLaMA-3-8B-Instruct, KV Size = 128* | | | | | | | | | | | | | | |
| SLM | 26.21 | 21.09 | 43.69 | 29.56 | 15.82 | 16.21 | 41.33 | 74.01 | 33.13 | 12.67 | 77.33 | 56.66 | 48.40 | 38.16 |
| +CaliDrop | 30.89 | 26.28 | 47.56 | 31.20 | 17.98 | 19.41 | 61.33 | 84.39 | 34.90 | 12.33 | 77.67 | 56.75 | 46.49 | **42.09** |
| H2O | 36.17 | 25.48 | 48.65 | 31.74 | 20.76 | 20.67 | 40.00 | 87.30 | 36.04 | 12.33 | 84.00 | 58.60 | 51.23 | 42.54 |
| +CaliDrop | 34.90 | 29.42 | 49.91 | 32.62 | 22.43 | 21.66 | 62.00 | 89.42 | 37.60 | 13.17 | 83.67 | 56.55 | 49.46 | **44.83** |
| SnapKV | 37.76 | 29.75 | 49.19 | 32.13 | 19.89 | 20.18 | 47.33 | 87.86 | 35.33 | 11.11 | 82.33 | 60.29 | 50.87 | 43.39 |
| +CaliDrop | 37.23 | 30.87 | 50.45 | 33.19 | 21.67 | 21.57 | 65.00 | 89.35 | 36.94 | 12.44 | 83.00 | 56.19 | 49.07 | **45.15** |

**LongBench** (Bai et al., 2024): This benchmark is a comprehensive benchmark to evaluate the contextual understanding capabilities of LLMs. It includes tasks such as answering questions, summarizing, and generating code.

**RULER** (Hsieh et al., 2024): RULER is a benchmark to evaluate the long-context modeling capabilities of LLMS. It involves tasks that require understanding and integrating various pieces of information, making it essential for assessing skills in complex multi-hop and aggregation tasks.

**Needle-in-a-Haystack** (Liu et al., 2024c): This benchmark focuses on testing if models can find important details in long texts. It checks how well models can spot useful information in a lot of text, which is key for tasks like finding facts or answering questions by pulling out parts of the text.

### 4.1.2 BASELINES

We evaluate the performance improvement of CaliDrop on top of three baselines.

**StreamingLLM (SLM)** (Xiao et al., 2024) identifies the *attention sink* phenomenon and enables LLMs to process infinitely long texts by maintaining KV cache of the attention sink along with a sliding window of recent tokens.

**Heavy Hitter Oracle (H2O)** (Zhang et al., 2023) measures the importance of tokens using accumulated attention scores. It retains a sliding window along with the KV cache of important tokens, thereby preserving the most useful information.

**SnapKV** (Li et al., 2024) achieves KV cache compression by selecting clustered important tokens and utilizing a local window in prefilling phase. It incorporates a clustering algorithm with a pooling layer and captures attention signals from an observation window.

**FullKV** caches all keys and values corresponding to each token, which is the standard approach for KV Cache in decoder-only transformer-based models.

**Note**: For simplicity, we follow the setting of SnapKV (Li et al., 2024) and perform compression only in the prefilling stage. Specifically, during prefilling, token eviction is conducted separately according to the strategies of each baselines. Afterward, the decoding process proceeds with our calibration for better accuracy. Although our approach only focuses on compression during the prefilling stage, it can be easily extended to the decoding stage as well.

### 4.1.3 Implementation Details

We evaluate the performance of various methods under different compression ratios by retaining {64, 128, 256, 512} tokens of the prompt. For StreamingLLM, we set the number of the attention sinks to 32, and the remaining tokens are allocated as local tokens. For H2O, we evenly split the token budget into important tokens and local tokens, each comprising half of the total. For SnapKV, we set the observation window size to 32 and employ average pooling with a kernel size of 5 for importance evaluation. For CaliDrop, we set $\theta_1$ to 0.7 and $\theta_2$ to 0.85, which achieves a trade-off between accuracy improvement and computational efficiency (see Section 5 for more details and exploration experiments). All experiments are conducted on NVIDIA A100 40G GPUs.

### 4.2 Main Results

#### 4.2.1 Results on LongBench

Table 1 and Appendix E show the main results of each method with different models in LongBench. We can conclude from the table that:

**CaliDrop can enhance the performance of various methods**: CaliDrop demonstrates significant performance improvements across a wide variety of tasks, effectively mitigating the performance degradation caused by KV cache compression. Specifically, in the Mistral-7B-Instruct and LLaMA-3-8B-Instruct models with a KV size of 64, CaliDrop consistently outperforms traditional baseline methods such as SLM, H2O, and SnapKV. This suggests that CaliDrop is highly effective in counteracting performance losses, especially under high compression ratios. Moreover, its performance gains are particularly notable in tasks that require multi-hop reasoning and multi-document summarization, highlighting its ability to maintain high performance even in complex and demanding scenarios.

**Marginal benefit of CaliDrop will diminish at higher KV sizes**: As the KV size approaches full capacity, the performance improvement brought by CaliDrop becomes increasingly constrained. This trend can be attributed to the fact that an already large KV cache provides ample information for the model to achieve satisfactory performance. Moreover, as shown in Equation 1, when the KV size increases, $\alpha_i$ will also increase, while $\alpha_j$ will decrease, reducing the calibration gain.

#### 4.2.2 Results on RULER

To further test our method in the long context of reasoning ability, we use the advanced long-context benchmark, RULER. In this experiment, we take Mistral-7B-Instruct and LLaMA-3-8B-Instruct as the base model and set the cache size to 64, 128, 256, and 512. As demonstrated in Appendix F, CaliDrop gains a comprehensive performance enhancement across different KV sizes. For example, with a KV size of 64, LLaMA-3-8B-Instruct using SnapKV with CaliDrop reaches **23.32%** compared to the baseline (**14.95%**). As the KV size increases, CaliDrop continues to outperform the baselines. On Ruler, we don't observe the gradual diminishing of the calibration gains as seen in LongBench. This is likely because the KV size of 512 was far from the required capacity for the task. Overall, the results demonstrate the effectiveness of CaliDrop.

#### 4.2.3 Results on Needle-in-a-Haystack

We also compare our method with baselines on Needle-in-a-Haystack. The results in Figure 2 and Appendix G indicate that CaliDrop achieves significant improvements based on token eviction methods across all settings in our experiment. Especially when using LLaMA-3-8B-Instruct with a KV size of 128, CaliDrop increases the accuracy of H2O from 48.5% to 83.8%. With a KV size of 256, when SnapKV is used in combination with CaliDrop, it manages to maintain almost no drop in performance in a context size of 8k, which shows the strong performance enhancement by CaliDrop.

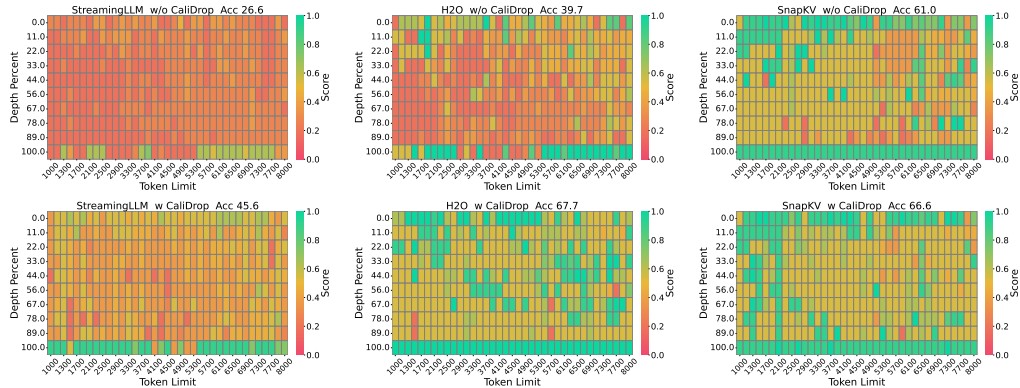

Figure 2: Results of Needle-in-a-Haystack on LLaMA-3-8B-Instruct with 8k context size and 64 KV size. The vertical axis of the figure represents the depth percentage, and the horizontal axis represents the token length.

Table 2: The influence on accuracy of the left threshold and the right threshold. The experiments are conducted on LongBench with LLaMA-3-8B-Instruct, SnapKV, and a KV size of 128.

| Group | | MF-en | Qasper | HotpotQA | 2WikiMQA | GovReport | MultiNews | TREC | TriviaQA | SAMSum | PCount | PRe | Lcc | RB-P | AVG |
|---|---|---|---|---|---|---|---|---|---|---|---|---|---|---|---|
| **R=0.80** | L=0.60 | 36.80 | 30.28 | 49.46 | 32.78 | 20.70 | 20.80 | 61.67 | 88.62 | 36.21 | 13.28 | 82.00 | 61.33 | 51.24 | 45.01 |
| | L=0.65 | 37.63 | 31.67 | 49.73 | 32.74 | 21.14 | 21.19 | 63.67 | 88.61 | 37.13 | 12.89 | 83.33 | 58.09 | 50.37 | 45.25 |
| | L=0.70 | 37.99 | 30.80 | 49.53 | 32.66 | 22.26 | 21.81 | 65.33 | 88.95 | 37.60 | 12.11 | 83.00 | 58.08 | 49.67 | 45.37 |
| | L=0.75 | 38.53 | 33.60 | 48.73 | 32.05 | 22.79 | 22.11 | 65.33 | 89.01 | 38.83 | 10.78 | 83.00 | 58.60 | 49.42 | **45.60** |
| | L=0.80 | 38.66 | 32.71 | 48.85 | 31.88 | 23.77 | 22.79 | 65.67 | 88.64 | 38.29 | 12.00 | 82.67 | 57.39 | 48.38 | 45.52 |
| **R=0.85** | L=0.60 | 37.98 | 31.74 | 49.05 | 32.71 | 20.50 | 20.64 | 59.67 | 88.77 | 36.31 | 12.22 | 82.33 | 60.65 | 51.24 | 44.91 |
| | L=0.65 | 36.94 | 31.43 | 49.64 | 33.16 | 20.91 | 21.08 | 62.67 | 88.75 | 36.38 | 12.44 | 83.33 | 57.81 | 50.88 | 45.03 |
| | L=0.70 | 37.23 | 30.87 | 50.45 | 33.19 | 21.67 | 21.57 | 65.00 | 89.35 | 36.94 | 12.44 | 83.00 | 56.19 | 49.07 | 45.15 |
| | L=0.75 | 37.84 | 32.47 | 49.78 | 34.12 | 22.61 | 22.09 | 64.33 | 89.25 | 37.65 | 11.78 | 83.33 | 58.70 | 51.34 | 45.79 |
| | L=0.80 | 38.94 | 33.85 | 49.74 | 34.15 | 24.19 | 22.80 | 64.33 | 89.53 | 38.74 | 11.44 | 84.00 | 59.11 | 51.22 | **46.31** |
| **R=0.90** | L=0.60 | 37.51 | 30.88 | 49.00 | 33.47 | 20.49 | 20.52 | 55.33 | 88.54 | 35.85 | 13.11 | 83.00 | 60.17 | 52.70 | 44.66 |
| | L=0.65 | 38.10 | 31.47 | 49.12 | 33.68 | 20.56 | 20.82 | 59.00 | 87.90 | 36.04 | 12.44 | 82.67 | 58.73 | 51.86 | 44.80 |
| | L=0.70 | 39.06 | 31.24 | 50.51 | 33.45 | 21.34 | 21.35 | 59.67 | 89.06 | 36.18 | 12.22 | 83.00 | 58.70 | 51.79 | 45.20 |
| | L=0.75 | 37.77 | 32.84 | 49.77 | 33.52 | 21.89 | 21.85 | 58.33 | 89.65 | 36.93 | 13.78 | 83.33 | 59.91 | 52.18 | 45.52 |
| | L=0.80 | 40.46 | 33.57 | 50.17 | 34.83 | 23.17 | 22.06 | 58.67 | 90.03 | 37.75 | 11.78 | 83.33 | 58.90 | 52.52 | **45.94** |

## 5 DISCUSSION

CaliDrop has two key hyperparameters: $\theta_1$ and $\theta_2$. Specifically, $\theta_1$ controls the frequency of recomputations, while $\theta_2$ determines the threshold conditions under which calibration is deemed acceptable. These hyperparameters directly influence the model throughput and accuracy. Throughput is primarily affected by $\theta_1$, as it is directly tied to the frequency of recomputations. However, accuracy is influenced by both $\theta_1$ and $\theta_2$. $\theta_1$ affects calibration quality, with more frequent recomputations leading to better calibration. $\theta_2$ determines the criteria for when calibration is considered acceptable. If the conditions are too strict, the model may not fully utilize the benefits of calibration. On the other hand, if the conditions are too lenient, the model may suffer from low-quality calibration, which could reduce accuracy. Therefore, in this section, we focus on how to improve accuracy while minimizing the decrease in model throughput.

### 5.1 INFLUENCE ON ACCURACY

Table 2 shows the influence on accuracy of $\theta_1$ and $\theta_2$. We can conclude that, with a fixed $\theta_2$, higher $\theta_1$ lead to increased accuracy, which aligns with our hypothesis. This suggests that by increasing the frequency of recomputations, the quality of calibration improves, thereby enhancing accuracy. When $\theta_1$ is fixed and $\theta_2$ is varied, accuracy does not follow a consistent trend. This occurs because a high $\theta_2$ can discard some effective calibration, while a low $\theta_2$ may introduce potentially harmful calibration. This behavior is also consistent with our hypothesis. Based on these results of accuracy and other explorations of efficiency, we set $\theta_1 = 0.7$ and $\theta_2 = 0.85$ for our main experiment.

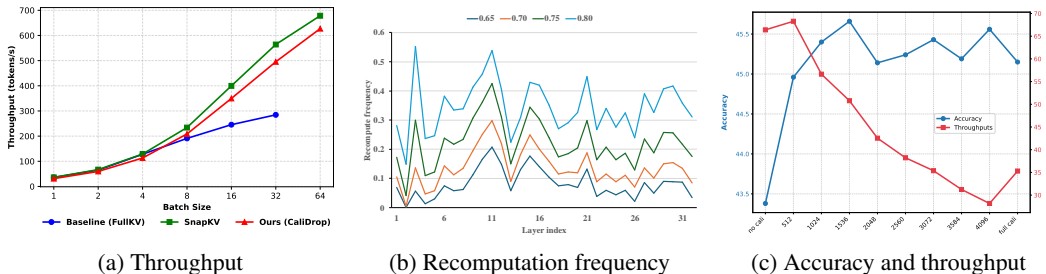

(a) Throughput      (b) Recomputation frequency      (c) Accuracy and throughput

Figure 3: (a) Throughput of CaliDrop and SnapKV under different batch size. (b) Recomputation frequency of CaliDrop in different layers under different $\theta_1$.(c) Accuracy and throughput under different calibration sizes. No cali: without calibration. Full cali: all evicted tokens are used for calibration (calibration size $= \infty$). We set $\theta_1 = 0.7$ in our experiment.

## 5.2 SPEED ANALYSIS

We evaluate throughput across varying batch sizes using an input length of 1024, output length of 128, and a KV cache budget of 128 on an 80GB A800. As shown in Figure 3a, SnapKV achieves notable throughput gains over FullKV due to its reduced memory footprint. CaliDrop adds only marginal computational overhead compared to SnapKV, while maintaining high throughput across practical batch sizes. Notably, its calibration-based accuracy improvement incurs minimal latency impact, showing that CaliDrop effectively balances efficiency and performance.

We also investigate the frequency of recomputation. We conduct experiments on LongBench using LLaMA-3-8B-Instruct with SnapKV and a KV size of 128. We compare the recompute frequency (the ratio of recomputations to total decoding steps) in different layers under different $\theta_1$ in Figure 3b. We find that the degree of variation in query similarity between different layers is inconsistent, leading to differences in recompute frequencies across layers. Additionally, as observed earlier, a higher $\theta_1$ leads to more recomputations. Overall, in our main experiment ($\theta_1$=0.7), recomputation occurs approximately every eight steps.

## 5.3 EFFICIENCY OPTIMIZATION

In our main experiment, we did not apply any efficiency optimizations. However, various techniques can be used to improve efficiency. In principle, most existing KV cache optimization methods—such as quantization and token eviction—are applicable to the offloaded cache. Additional strategies like specialized CUDA kernels and communication overlapping may also benefit CaliDrop. Since the theoretical computation overhead is low (recomputation occurs approximately every 8 steps), we leave extensive efficiency optimization to future work. As a lightweight yet effective enhancement, we introduce a hyper-parameter called calibration size, which controls the number of offloaded tokens by ranking them according to importance. This approach significantly reduces calibration overhead while preserving most of the accuracy benefits. Figure 3c shows the accuracy of LLaMA-3-8B-Instruct using SnapKV and CaliDrop on LongBench under different calibration sizes, along with throughput results from the experiment setup in Section 5.2. We find that a small calibration size can lead to significant performance improvements with only a modest sacrifice in throughput.

## 6 CONCLUSION

In this work, we propose CaliDrop, a strategy that enhances token eviction methods at the cost of lightweight additional computations. We leverage the similarity between queries to add precomputed calibration for future queries, and offload the evicted tokens for future recomputation. Calidrop effectively improves accuracy based on token eviction methods, allowing them to perform better in scenarios with lower compression ratios. We also explore a series of efficiency-related issues concerning CaliDrop and propose several potential optimization solutions.

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

## A  ETHICS STATEMENT

We confirm that this work adheres to ethical research practices. All data and LLMs used are publicly available (including API format) and properly cited. No human subjects were involved.

## B  REPRODUCIBILITY STATEMENT

Comprehensive details of the experimental settings, hyperparameters, and evaluation protocols are presented in Section 4.1. The implementation, including source code and execution scripts, will be made publicly available upon publication.Experiments are conducted on NVIDIA GPUs with PyTorch, HuggingFace Transformers (Wolf et al., 2020)

## C  USE OF LLMs

During the writing of this paper, we leverage large language models (LLMs) to refine the clarity and fluency of our writing, particularly in the Abstract and Introduction sections. Specifically, we used the Qwen web interface [2] to access the Qwen series of models (e.g., Qwen-Max), inputting early drafts of these sections and requesting stylistic improvements while preserving technical accuracy and original intent. The model's suggestions helped enhance sentence structure, academic tone, and overall readability. All final content was carefully reviewed, validated, and edited by the authors to ensure fidelity to our research and adherence to scholarly standards.

## D  TASK ABBREVIATIONS

For clarity, we provide a mapping between the abbreviated task names used in tables and figures and their full descriptions :

Table 3: Mapping of task abbreviations to full names.

| Abbr. | Full Name |
|---|---|
| MF-en | MultiFieldQA_en |
| PCount | Passage_Count |
| PRe | Passage_Retrieval_en |
| RB-P | Repobench-P |
| S-NIAH-1 | Single Needle-in-a-Haystack 1 |
| S-NIAH-2 | Single Needle-in-a-Haystack 2 |
| S-NIAH-3 | Single Needle-in-a-Haystack 3 |
| MK-NIAH-1 | Multi-key Needle-in-a-Haystack 1 |
| MK-NIAH-2 | Multi-key Needle-in-a-Haystack 2 |
| MK-NIAH-3 | Multi-key Needle-in-a-Haystack 3 |
| MV-NIAH | Multi-values Needle-in-a-Haystack |
| MQ-NIAH | Multi-queries Needle-in-a-Haystack |
| VT | Variable Tracking |
| CWE | Common Words Extraction |
| FWE | Frequent Words Extraction |

## E  MORE RESULTS ON LONGBENCH.

Table 4 presents a comprehensive comparison of CaliDrop when applied to various token eviction baselines—SLM, H2O, and SnapKV—across multiple tasks in the LongBench benchmark. Results are reported for both LLaMA-3-8B-Instruct and Mistral-7B-Instruct under different KV cache budgets (256, 512) and full KV settings.

---

[2]https://chat.qwen.ai

Table 4: Performance comparison of CaliDrop with SnapKV, H2O, StreamingLLM (SLM) and FullKV on Long- Bench for LLaMA-3-8B-Instruct, Mistral-7B-Instruct and LLaMA-3-70B-Instruct. CaliDrop generally achieves improvements over previous KV cache compression methods across various KV cache sizes and LLMs. Bold text represents the best performance.

| Method | Single-Document QA | | Multi-Document QA | | Summarization | | Few-shot Learning | | | Synthetic | | Code | | Avg. |
|---|---|---|---|---|---|---|---|---|---|---|---|---|---|---|
| | MF-en | Qasper | HotpotQA | 2WikiMQA | GovReport | MultiNews | TREC | TriviaQA | SAMSum | PCount | PRe | Lcc | RB-P | |
| | 18409 | 3619 | 9151 | 4887 | 8734 | 2113 | 5177 | 8209 | 6258 | 11141 | 9289 | 1235 | 4206 | |
| *Mistral-7B-Instruct, KV Size = Full* | | | | | | | | | | | | | | |
| FullKV | 48.54 | 24.35 | 32.92 | 21.87 | 33.05 | 25.77 | 67.00 | 86.84 | 40.95 | 5.40 | 91.00 | 57.24 | 49.84 | 44.98 |
| *Mistral-7B-Instruct, KV Size = 256* | | | | | | | | | | | | | | |
| SLM | 24.00 | 9.30 | 17.54 | 14.58 | 17.21 | 16.87 | 48.33 | 79.98 | 38.11 | 4.47 | 42.48 | 52.50 | 40.97 | 31.26 |
| +CaliDrop | 29.26 | 13.04 | 22.63 | 16.25 | 21.91 | 21.60 | 54.67 | 80.06 | 38.29 | 6.24 | 59.63 | 53.58 | 43.33 | **35.42** |
| H2O | 34.18 | 14.97 | 19.83 | 15.66 | 22.54 | 21.93 | 48.00 | 85.54 | 39.01 | 5.99 | 83.40 | 54.67 | 45.42 | 37.78 |
| +CaliDrop | 39.53 | 18.27 | 24.62 | 17.54 | 24.74 | 23.17 | 58.33 | 85.35 | 39.31 | 6.06 | 85.70 | 53.05 | 47.17 | **40.22** |
| SnapKV | 42.87 | 17.33 | 24.65 | 16.65 | 22.78 | 22.23 | 58.00 | 86.77 | 38.67 | 6.40 | 90.57 | 54.91 | 45.69 | 40.58 |
| +CaliDrop | 43.97 | 22.32 | 26.46 | 18.66 | 24.94 | 23.11 | 57.33 | 86.78 | 39.23 | 6.00 | 90.08 | 54.56 | 46.97 | **41.57** |
| *Mistral-7B-Instruct, KV Size = 512* | | | | | | | | | | | | | | |
| SLM | 24.97 | 10.47 | 18.02 | 14.76 | 21.20 | 20.03 | 53.67 | 82.12 | 39.07 | 5.47 | 42.49 | 55.37 | 43.13 | 33.14 |
| +CaliDrop | 31.47 | 13.97 | 24.15 | 16.18 | 23.99 | 22.49 | 55.33 | 82.70 | 39.38 | 6.67 | 59.86 | 55.11 | 45.59 | **36.68** |
| H2O | 36.32 | 16.50 | 21.03 | 16.29 | 24.19 | 23.24 | 53.67 | 86.58 | 39.52 | 6.19 | 86.37 | 55.7 | 47.09 | 39.44 |
| +CaliDrop | 41.43 | 20.29 | 26.27 | 17.73 | 25.90 | 24.13 | 56.67 | 86.56 | 39.83 | 6.63 | 86.88 | 55.42 | 46.83 | **41.12** |
| SnapKV | 45.24 | 20.09 | 26.64 | 18.27 | 25.18 | 23.25 | 62.67 | 86.51 | 39.45 | 5.73 | 92.22 | 57.09 | 47.85 | 42.32 |
| +CaliDrop | 45.03 | 22.27 | 29.66 | 20.73 | 26.60 | 23.81 | 59.00 | 86.86 | 39.21 | 5.77 | 92.33 | 55.56 | 47.84 | **42.67** |
| *LLaMA-3-8B-Instruct, KV Size = Full* | | | | | | | | | | | | | | |
| FullKV | 40.56 | 37.54 | 49.81 | 34.93 | 31.04 | 25.58 | 69.67 | 89.85 | 40.50 | 12.94 | 83.67 | 56.58 | 51.01 | 47.98 |
| *LLaMA-3-8B-Instruct, KV Size = 256* | | | | | | | | | | | | | | |
| SLM | 26.78 | 22.31 | 42.52 | 29.17 | 18.51 | 19.29 | 50.67 | 80.23 | 36.52 | 15.00 | 78.33 | 58.52 | 51.56 | 40.72 |
| +CaliDrop | 33.11 | 25.70 | 47.92 | 31.40 | 20.06 | 21.31 | 63.33 | 84.59 | 37.44 | 12.69 | 78.67 | 57.55 | 48.47 | **43.25** |
| H2O | 35.78 | 28.26 | 48.71 | 31.51 | 21.78 | 21.44 | 45.33 | 89.45 | 37.92 | 12.11 | 83.67 | 60.02 | 51.56 | 43.66 |
| +CaliDrop | 36.04 | 30.67 | 49.36 | 33.27 | 23.03 | 22.60 | 63.00 | 90.06 | 38.69 | 12.94 | 83.67 | 58.03 | 52.24 | **45.66** |
| SnapKV | 38.45 | 30.77 | 49.70 | 33.80 | 22.15 | 21.71 | 57.00 | 89.28 | 36.79 | 12.11 | 84.00 | 60.22 | 53.11 | 45.31 |
| +CaliDrop | 39.11 | 32.72 | 50.08 | 33.63 | 23.49 | 22.54 | 66.00 | 89.99 | 38.04 | 13.33 | 83.67 | 56.57 | 51.36 | **46.19** |
| *LLaMA-3-8B-Instruct, KV Size = 512* | | | | | | | | | | | | | | |
| SLM | 27.12 | 22.83 | 43.69 | 29.94 | 21.90 | 21.94 | 56.67 | 82.81 | 37.81 | 14.31 | 76.33 | 60.62 | 53.25 | 42.25 |
| +CaliDrop | 33.05 | 27.57 | 47.96 | 32.54 | 22.70 | 22.85 | 65.67 | 86.73 | 38.12 | 12.48 | 78.67 | 58.26 | 49.00 | **44.28** |
| H2O | 37.64 | 30.66 | 50.20 | 32.93 | 23.54 | 22.91 | 54.00 | 89.70 | 38.89 | 10.61 | 83.67 | 59.10 | 52.59 | 45.11 |
| +CaliDrop | 38.17 | 32.68 | 49.77 | 33.43 | 24.63 | 23.42 | 64.67 | 90.06 | 39.08 | 11.50 | 83.67 | 59.77 | 51.93 | **46.37** |
| SnapKV | 38.34 | 33.65 | 50.66 | 34.62 | 24.48 | 23.06 | 65.67 | 90.03 | 38.47 | 12.56 | 83.67 | 59.90 | 52.02 | 46.70 |
| +CaliDrop | 39.30 | 34.99 | 50.23 | 34.86 | 25.21 | 23.47 | 67.00 | 90.14 | 38.58 | 13.11 | 84.00 | 59.28 | 52.42 | **47.12** |
| *LLaMA-3-70B-Instruct, KV Size = Full* | | | | | | | | | | | | | | |
| FullKV | 47.44 | 44.15 | 62.12 | 54.97 | 31.52 | 25.21 | 71.33 | 91.24 | 43.56 | 22.33 | 89.33 | 61.71 | 63.30 | 54.48 |
| *LLaMA-3-70B-Instruct, KV Size = 64* | | | | | | | | | | | | | | |
| SLM | 30.69 | 24.51 | 53.82 | 50.26 | 15.28 | 14.58 | 38.00 | 86.66 | 28.53 | 22.67 | 89.00 | 53.77 | 46.60 | 42.64 |
| +CaliDrop | 39.31 | 31.52 | 54.58 | 50.95 | 17.85 | 18.45 | 59.67 | 85.03 | 30.83 | 22.67 | 89.00 | 50.82 | 50.45 | **46.24** |
| H2O | 42.98 | 30.01 | 58.08 | 52.90 | 20.45 | 20.87 | 42.00 | 89.74 | 38.49 | 22.44 | 88.67 | 56.72 | 54.55 | 47.53 |
| +CaliDrop | 44.06 | 35.00 | 58.12 | 52.11 | 21.51 | 21.53 | 62.00 | 90.55 | 39.65 | 22.44 | 88.67 | 58.37 | 55.91 | **49.99** |
| SnapKV | 42.35 | 31.78 | 58.51 | 53.62 | 18.44 | 19.15 | 44.00 | 89.41 | 37.12 | 22.67 | 88.33 | 57.42 | 54.10 | 47.45 |
| +CaliDrop | 43.23 | 34.60 | 58.22 | 51.73 | 20.03 | 20.52 | 64.00 | 91.18 | 38.91 | 23.00 | 88.67 | 56.90 | 54.59 | **49.66** |

The results demonstrate that CaliDrop consistently enhances the performance of all baselines, with particularly notable improvements in low-budget settings (e.g., KV size = 64). For instance, on LLaMA-3-70B-Instruct with a 64-token cache, CaliDrop boosts H2O's average score from 47.53 to 49.99 and SnapKV's from 47.45 to 49.66. The gains are more pronounced in *Few-shot Learning* and *Summarization*, where accurate attention over long-range dependencies is crucial. As the KV budget increases, the relative improvement stabilizes.

# F   MORE RESULTS ON RULER

Tables 5 and 6 present a comprehensive evaluation of CaliDrop on the RULER benchmark, which assesses a model's ability to retrieve critical information across extremely long sequences (up to 32k tokens) under varying compression ratios. We systematically evaluate both LLaMA-3-8B-Instruct and Mistral-7B-Instruct across multiple KV cache sizes (64, 128, 256, 512), and compare CaliDrop-enhanced variants of SLM, H2O, SnapKV against their respective baselines and FullKV.

As shown in Tables 5 and 6, CaliDrop consistently improves the needle-retrieval accuracy of all base-line methods across both models and all compression levels. The gains are particularly pronounced under aggressive compression (e.g., KV size = 64), where standard token eviction strategies struggle to preserve essential context. For instance, on LLaMA-3-8B-Instruct with a 64-token cache, SnapKV achieves an average accuracy of 14.95%, while `SnapKV + CaliDrop` boosts this to **23.32%**—a relative improvement of over 56%. Similarly, for Mistral-7B-Instruct at the same budget, CaliDrop

Table 5: Performance comparison of CaliDrop with SnapKV, H2O, StreamingLLM (SLM) and FullKV on RULER for LLaMA-3-8B-Instruct. Bold text represents the best performance.

| Method | Single NIAH | | | Multi-key NIAH | | | MQ-NIAH | MV-NIAH | CWE | FWE | VT | Avg. |
|---|---|---|---|---|---|---|---|---|---|---|---|---|
| | S-NIAH-1 | S-NIAH-2 | S-NIAH-3 | MK-NIAH-1 | MK-NIAH-2 | MK-NIAH-3 | | | | | | |
| *LLaMA-3-8B-Instruct, KV Size = Full* | | | | | | | | | | | | |
| FullKV | 100.00 | 98.20 | 97.00 | 99.20 | 91.60 | 95.80 | 99.75 | 97.45 | 97.82 | 82.27 | 98.28 | 96.12 |
| *LLaMA-3-8B-Instruct, KV Size = 64* | | | | | | | | | | | | |
| SLM | 0 | 0 | 0 | 0 | 0 | 0 | 0 | 0 | 0.12 | 28.80 | 0 | 2.63 |
| +CaliDrop | 1.60 | 1.20 | 0 | 2.20 | 0.40 | 0 | 2.20 | 1.80 | 0.58 | 31.33 | 4.00 | **4.12** |
| H2O | 20.40 | 20.80 | 0 | 5.40 | 0.40 | 0 | 0.20 | 0.15 | 4.62 | 0.20 | 2.96 | 5.01 |
| +CaliDrop | 76.60 | 40.20 | 0 | 17.00 | 7.80 | 0 | 15.30 | 14.50 | 12.16 | 3.60 | 11.24 | 18.04 |
| SnapKV | 58.40 | 67.20 | 0 | 21.40 | 13.80 | 0 | 0.75 | 0.35 | 0.16 | 0.20 | 2.24 | 14.95 |
| +CaliDrop | 77.60 | 68.20 | 0 | 40.20 | 32.00 | 0 | 12.85 | 12.70 | 1.18 | 2.07 | 9.68 | **23.32** |
| *LLaMA-3-8B-Instruct, KV Size = 128* | | | | | | | | | | | | |
| SLM | 0.40 | 1.20 | 2.40 | 3.00 | 0.40 | 0 | 2.45 | 2.45 | 6.84 | 52.20 | 0.64 | 6.54 |
| +CaliDrop | 2.40 | 2.80 | 2.40 | 3.00 | 0.60 | 0 | 5.00 | 4.00 | 13.72 | 51.27 | 5.96 | **8.29** |
| H2O | 41.00 | 43.80 | 2.40 | 15.00 | 2.40 | 0 | 2.30 | 0.30 | 18.00 | 15.13 | 7.96 | 13.48 |
| +CaliDrop | 73.60 | 61.00 | 2.40 | 26.20 | 15.60 | 0 | 30.85 | 22.60 | 36.94 | 23.87 | 16.80 | 28.17 |
| SnapKV | 98.80 | 87.00 | 0 | 60.80 | 33.80 | 0 | 17.15 | 7.45 | 4.60 | 31.33 | 13.20 | 32.19 |
| +CaliDrop | 99.00 | 87.20 | 0 | 65.80 | 45.40 | 0 | 49.40 | 40.50 | 13.94 | 39.07 | 24.28 | **42.24** |
| *LLaMA-3-8B-Instruct, KV Size = 256* | | | | | | | | | | | | |
| SLM | 1.40 | 1.20 | 2.40 | 3.00 | 1.40 | 1.20 | 2.45 | 2.45 | 17.42 | 75.60 | 3.12 | 10.15 |
| +CaliDrop | 2.80 | 2.80 | 2.40 | 4.40 | 1.60 | 1.20 | 5.40 | 4.05 | 25.48 | 70.53 | 8.48 | **11.74** |
| H2O | 66.60 | 56.60 | 2.40 | 25.00 | 15.20 | 0 | 8.90 | 1.30 | 31.64 | 49.93 | 20.68 | 25.30 |
| +CaliDrop | 83.20 | 68.00 | 2.40 | 40.60 | 20.80 | 0 | 42.70 | 29.80 | 58.18 | 55.20 | 38.76 | 39.97 |
| SnapKV | 100.00 | 92.60 | 0 | 90.20 | 26.00 | 0 | 76.40 | 36.50 | 13.40 | 47.67 | 91.48 | 52.20 |
| +CaliDrop | 100.00 | 93.20 | 0 | 89.80 | 36.60 | 0 | 87.15 | 73.90 | 40.90 | 57.33 | 94.08 | **61.18** |
| *LLaMA-3-8B-Instruct, KV Size = 512* | | | | | | | | | | | | |
| SLM | 4.00 | 6.20 | 8.00 | 7.00 | 5.20 | 3.80 | 5.25 | 6.80 | 35.66 | 74.00 | 7.68 | 14.87 |
| +CaliDrop | 6.20 | 7.40 | 8.00 | 8.60 | 5.60 | 3.80 | 8.20 | 8.00 | 41.38 | 75.67 | 12.64 | **16.86** |
| H2O | 88.20 | 65.60 | 2.40 | 44.00 | 25.60 | 1.40 | 24.30 | 3.00 | 46.08 | 64.33 | 69.28 | 39.47 |
| +CaliDrop | 93.80 | 75.80 | 2.40 | 53.80 | 23.40 | 1.40 | 59.45 | 39.50 | 75.18 | 70.07 | 81.84 | 52.42 |
| SnapKV | 100.00 | 93.60 | 1.60 | 96.80 | 31.00 | 0.20 | 91.10 | 71.65 | 30.50 | 56.20 | 94.36 | 60.64 |
| +CaliDrop | 100.00 | 94.20 | 1.40 | 97.20 | 37.40 | 0.20 | 95.55 | 86.50 | 67.24 | 67.47 | 96.40 | **67.60** |
| *Mistral-7B-Instruct, KV Size = Full* | | | | | | | | | | | | |
| FullKV | 100.00 | 100.00 | 96.40 | 94.60 | 99.00 | 80.60 | 95.75 | 92.85 | 83.72 | 83.93 | 96.48 | 93.03 |
| *Mistral-7B-Instruct, KV Size = 64* | | | | | | | | | | | | |
| SLM | 0 | 0 | 0 | 0 | 0 | 0 | 0 | 0 | 0.28 | 6.13 | 0 | 0.58 |
| +CaliDrop | 0.20 | 0 | 0 | 0 | 0 | 0 | 0 | 0 | 1.14 | 8.20 | 2.32 | **1.08** |
| H2O | 0 | 0 | 0 | 0 | 0 | 0 | 0 | 0 | 3.50 | 0.07 | 5.00 | 0.78 |
| +CaliDrop | 6.60 | 1.80 | 0 | 0.20 | 0.20 | 0 | 0.10 | 0.15 | 12.30 | 2.27 | 12.92 | 3.32 |
| SnapKV | 0.20 | 0.40 | 0 | 0.20 | 0 | 0 | 0 | 0 | 0.80 | 0.07 | 3.20 | 0.44 |
| +CaliDrop | 9.20 | 4.40 | 0 | 2.60 | 1.40 | 0 | 0.10 | 0.15 | 6.38 | 1.20 | 8.60 | **3.09** |
| *Mistral-7B-Instruct, KV Size = 128* | | | | | | | | | | | | |
| SLM | 0.40 | 1.20 | 2.20 | 3.00 | 0.40 | 0 | 2.40 | 2.25 | 4.26 | 11.47 | 0.56 | 2.56 |
| +CaliDrop | 0.40 | 1.20 | 2.20 | 3.00 | 0.40 | 0 | 2.15 | 2.00 | 26.96 | 12.93 | 2.96 | **4.93** |
| H2O | 2.20 | 2.20 | 0.20 | 3.20 | 0.20 | 0 | 0 | 0.30 | 14.40 | 13.47 | 11.40 | 4.32 |
| +CaliDrop | 13.20 | 6.40 | 0.80 | 4.40 | 0.60 | 0 | 0.05 | 1.30 | 23.10 | 12.47 | 25.16 | 7.95 |
| SnapKV | 64.20 | 23.00 | 0 | 8.80 | 4.00 | 0 | 0.05 | 0.05 | 2.68 | 41.20 | 16.36 | 14.58 |
| +CaliDrop | 79.40 | 40.00 | 0 | 17.60 | 8.20 | 0 | 0.60 | 3.15 | 8.26 | 43.47 | 27.72 | **20.76** |
| *Mistral-7B-Instruct, KV Size = 256* | | | | | | | | | | | | |
| SLM | 1.40 | 1.20 | 2.20 | 3.00 | 1.00 | 0.60 | 2.35 | 2.30 | 9.56 | 29.93 | 2.40 | 5.09 |
| +CaliDrop | 1.40 | 1.20 | 2.20 | 3.00 | 1.00 | 0.60 | 2.25 | 1.85 | 34.82 | 32.27 | 4.60 | **7.74** |
| H2O | 29.40 | 5.80 | 2.00 | 4.20 | 0.40 | 0 | 1.45 | 0.50 | 30.00 | 46.87 | 22.72 | 13.03 |
| +CaliDrop | 44.80 | 16.80 | 2.20 | 7.80 | 1.00 | 0 | 1.85 | 1.80 | 38.90 | 36.07 | 42.12 | 17.58 |
| SnapKV | 93.60 | 54.40 | 0 | 35.60 | 8.60 | 0 | 1.55 | 0.65 | 8.54 | 62.60 | 80.64 | 31.47 |
| +CaliDrop | 95.20 | 59.80 | 0.20 | 43.60 | 16.20 | 0 | 8.50 | 8.00 | 23.22 | 70.27 | 84.24 | **37.20** |
| *Mistral-7B-Instruct, KV Size = 512* | | | | | | | | | | | | |
| SLM | 3.80 | 5.60 | 5.60 | 5.40 | 4.00 | 3.00 | 5.05 | 4.40 | 18.94 | 58.80 | 7.04 | 11.06 |
| +CaliDrop | 3.80 | 5.60 | 5.60 | 5.40 | 4.00 | 3.00 | 5.00 | 4.00 | 39.12 | 53.93 | 8.28 | **12.52** |
| H2O | 74.80 | 21.00 | 2.00 | 9.60 | 2.20 | 0.60 | 1.75 | 0.50 | 37.66 | 65.87 | 49.24 | 24.11 |
| +CaliDrop | 80.40 | 33.00 | 1.80 | 16.60 | 4.40 | 0.60 | 3.80 | 1.80 | 49.04 | 64.20 | 67.00 | 29.33 |
| SnapKV | 98.00 | 75.80 | 0.60 | 57.40 | 20.8 | 0 | 10.45 | 3.60 | 17.36 | 71.33 | 92.68 | 40.73 |
| +CaliDrop | 98.00 | 77.20 | 2.80 | 60.40 | 28.40 | 0.60 | 28.50 | 13.20 | 40.78 | 78.67 | 92.60 | **47.38** |

elevates SnapKV from 0.44% to **3.09%**, demonstrating its critical role in enabling basic retrieval capability under extreme memory constraints.

The performance advantage of CaliDrop persists across higher KV budgets, with no signs of saturation even at 512 tokens. On Mistral-7B-Instruct, `SnapKV + CaliDrop` achieves an average accuracy of **47.38%**, surpassing the baseline by 6.65 percentage points. Notably, this gain is significantly larger than the marginal improvements observed in LongBench at comparable cache sizes, suggesting that RULER's more demanding reasoning and deep-context retrieval tasks benefit more substantially from CaliDrop's dynamic calibration mechanism.

Table 6: Performance comparison of CaliDrop with SnapKV, H2O, StreamingLLM (SLM) and FullKV on RULER for Mistral-7B-Instruct. The performance strengths of are more evident in small KV Cache sizes (i.e. KV Size = 64). Bold text represents the best performance.

| Method | Single NIAH | | | Multi-key NIAH | | | MQ-NIAH | MV-NIAH | CWE | FWE | VT | Avg. |
|---|---|---|---|---|---|---|---|---|---|---|---|---|
| | S-NIAH-1 | S-NIAH-2 | S-NIAH-3 | MK-NIAH-1 | MK-NIAH-2 | MK-NIAH-3 | | | | | | |
| Mistral-7B-Instruct, KV Size = Full | | | | | | | | | | | | |
| FullKV | 100.00 | 100.00 | 96.40 | 94.60 | 99.00 | 80.60 | 95.75 | 92.85 | 83.72 | 83.93 | 96.48 | 93.03 |
| Mistral-7B-Instruct, KV Size = 64 | | | | | | | | | | | | |
| SLM | 0 | 0 | 0 | 0 | 0 | 0 | 0 | 0 | 0.28 | 6.13 | 0 | 0.58 |
| +CaliDrop | 0.20 | 0 | 0 | 0 | 0 | 0 | 0 | 0 | 1.14 | 8.20 | 2.32 | **1.08** |
| H2O | 0 | 0 | 0 | 0 | 0 | 0 | 0 | 0 | 3.50 | 0.07 | 5.00 | 0.78 |
| +CaliDrop | 6.60 | 1.80 | 0 | 0.20 | 0.20 | 0 | 0.10 | 0.15 | 12.30 | 2.27 | 12.92 | **3.32** |
| SnapKV | 0.20 | 0.40 | 0 | 0.20 | 0 | 0 | 0 | 0 | 0.80 | 0.07 | 3.20 | 0.44 |
| +CaliDrop | 9.20 | 4.40 | 0 | 2.60 | 1.40 | 0 | 0.10 | 0.15 | 6.38 | 1.20 | 8.60 | **3.09** |
| Mistral-7B-Instruct, KV Size = 128 | | | | | | | | | | | | |
| SLM | 0.40 | 1.20 | 2.20 | 3.00 | 0.40 | 0 | 2.40 | 2.25 | 4.26 | 11.47 | 0.56 | 2.56 |
| +CaliDrop | 0.40 | 1.20 | 2.20 | 3.00 | 0.40 | 0 | 2.15 | 2.00 | 26.96 | 12.93 | 2.96 | **4.93** |
| H2O | 2.20 | 2.20 | 0.20 | 3.20 | 0.20 | 0 | 0 | 0.30 | 14.40 | 13.47 | 11.40 | 4.32 |
| +CaliDrop | 13.20 | 6.40 | 0.80 | 4.40 | 0.60 | 0 | 0.05 | 1.30 | 23.10 | 12.47 | 25.16 | **7.95** |
| SnapKV | 64.20 | 23.00 | 0 | 8.80 | 4.00 | 0 | 0.05 | 0.05 | 2.68 | 41.20 | 16.36 | 14.58 |
| +CaliDrop | 79.40 | 40.00 | 0 | 17.60 | 8.20 | 0 | 0.60 | 3.15 | 8.26 | 43.47 | 27.72 | **20.76** |
| Mistral-7B-Instruct, KV Size = 256 | | | | | | | | | | | | |
| SLM | 1.40 | 1.20 | 2.20 | 3.00 | 1.00 | 0.60 | 2.35 | 2.30 | 9.56 | 29.93 | 2.40 | 5.09 |
| +CaliDrop | 1.40 | 1.20 | 2.20 | 3.00 | 1.00 | 0.60 | 2.25 | 1.85 | 34.82 | 32.27 | 4.60 | **7.74** |
| H2O | 29.40 | 5.80 | 2.00 | 4.20 | 0.40 | 0 | 1.45 | 0.50 | 30.00 | 46.87 | 22.72 | 13.03 |
| +CaliDrop | 44.80 | 16.80 | 2.20 | 7.80 | 1.00 | 0 | 1.85 | 1.80 | 38.90 | 36.07 | 42.12 | **17.58** |
| SnapKV | 93.60 | 54.40 | 0 | 35.60 | 8.60 | 0 | 1.55 | 0.65 | 8.54 | 62.60 | 80.64 | 31.47 |
| +CaliDrop | 95.20 | 59.80 | 0.20 | 43.60 | 16.20 | 0 | 8.50 | 8.00 | 23.22 | 70.27 | 84.24 | **37.20** |
| Mistral-7B-Instruct, KV Size = 512 | | | | | | | | | | | | |
| SLM | 3.80 | 5.60 | 5.60 | 5.40 | 4.00 | 3.00 | 5.05 | 4.40 | 18.94 | 58.80 | 7.04 | 11.06 |
| +CaliDrop | 3.80 | 5.60 | 5.60 | 5.40 | 4.00 | 3.00 | 5.00 | 4.00 | 39.12 | 53.93 | 8.28 | **12.52** |
| H2O | 74.80 | 21.00 | 2.00 | 9.60 | 2.20 | 0.60 | 1.75 | 0.50 | 37.66 | 65.87 | 49.24 | 24.11 |
| +CaliDrop | 80.40 | 33.00 | 1.80 | 16.60 | 4.40 | 0.60 | 3.80 | 1.80 | 49.04 | 64.20 | 67.00 | **29.33** |
| SnapKV | 98.00 | 75.80 | 0.60 | 57.40 | 20.8 | 0 | 10.45 | 3.60 | 17.36 | 71.33 | 92.68 | 40.73 |
| +CaliDrop | 98.00 | 77.20 | 2.80 | 60.40 | 28.40 | 0.60 | 28.50 | 13.20 | 40.78 | 78.67 | 92.60 | **47.38** |

A closer examination of per-task performance reveals that CaliDrop provides the most substantial improvements in tasks involving multi-key and multi-value retrieval (e.g., MK-NIAH, MV-NIAH) and complex pattern recognition (e.g., VT, FWE). These are precisely the types of tasks where accurate attention over long-range dependencies is indispensable, and where naive eviction policies are most likely to discard semantically critical tokens. By leveraging query similarity to selectively reintroduce calibrated attention signals from previously evicted tokens, CaliDrop effectively mitigates information loss and significantly enhances the model's ability to reason over compressed contexts.

Furthermore, consistent gains across different architectures (LLaMA-3 and Mistral) confirm the generalizability of CaliDrop as a plug-in enhancement for token eviction methods. Unlike in LongBench, where performance gains diminish as the KV cache approaches full capacity, RULER's results show no such trend, indicating that the complexity and context length of the benchmark remain challenging even at moderate compression levels. This underscores the value of CaliDrop in real-world scenarios where models must operate under tight memory budgets while handling complex, long-term reasoning tasks.

## G MORE RESULTS ON NEEDLE-IN-A-HAYSTACK.

Figures 4, 5, 6, 7, 8, 9, 10, and 11 present the full evaluation results of Needle-in-a-Haystack for different models, context lengths, and KV cache sizes. These figures comprehensively visualize the retrieval accuracy as a function of the percentage of needle depth (vertical axis) and the length of the input sequence (horizontal axis), comparing the baseline methods (left column) with their CaliDrop-enhanced counterparts (right column).

CaliDrop consistently enhances the robustness of token eviction baselines across models and settings, with particularly significant gains in low-KV regimes (e.g., 64–128 tokens) where it effectively recovers lost attention signals through calibration, enabling reliable needle retrieval even at extreme depths—demonstrated by improvements from 48.5% to 83.8% for H2O and from 87.2% to 94.5% for SnapKV on LLaMA-3-8B-Instruct with 96-token cache. As the KV cache size increases, baseline performance approaches the full KV limit, leading to narrower relative gains, which aligns with

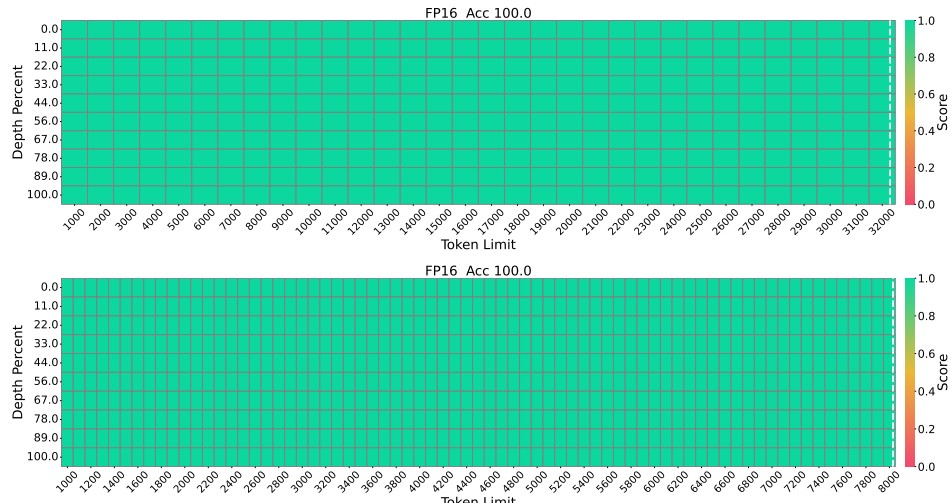

Figure 4: Results of Needle-in-a-Haystack on Mistral-7B-Instruct and LLaMA-3-8B-Instruct with 32k and 8k context size and full KV size. The vertical axis of the figure represents the depth percentage, and the horizontal axis represents the token length.

the method's upper-bound behavior. The consistent improvements on Mistral-7B-Instruct further confirm CaliDrop's generalizability across architectures, highlighting its effectiveness in preserving long-range attention patterns and enhancing the reliability of KV cache compression in practical long-context scenarios.

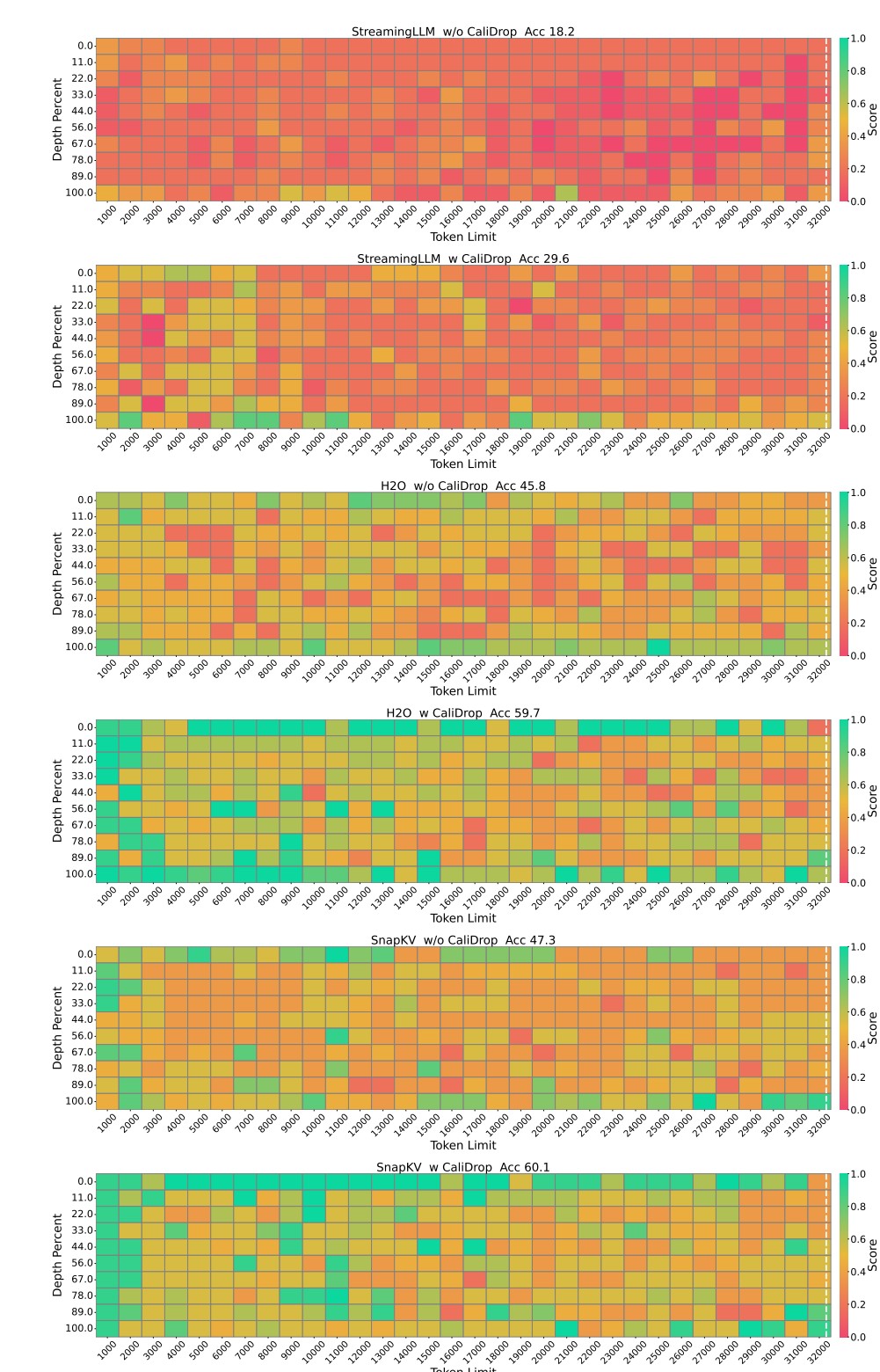

Figure 5: Results of Needle-in-a-Haystack on Mistral-7B-Instruct with 32k context size and 64 KV size. The vertical axis of the figure represents the depth percentage, and the horizontal axis represents the token length.

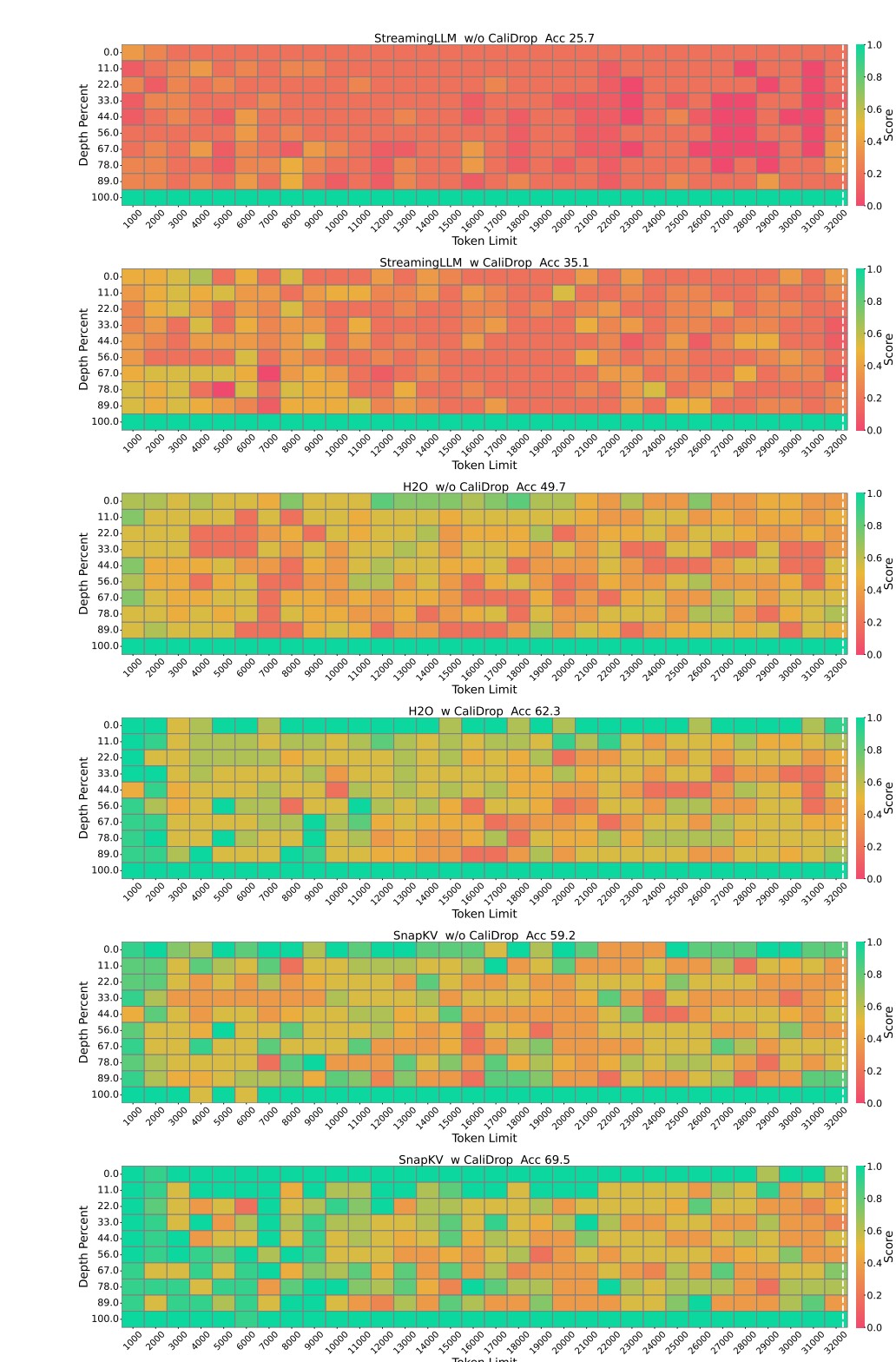

Figure 6: Results of Needle-in-a-Haystack on Mistral-7B-Instruct with 32k context size and 96 KV size. The vertical axis of the figure represents the depth percentage, and the horizontal axis represents the token length.

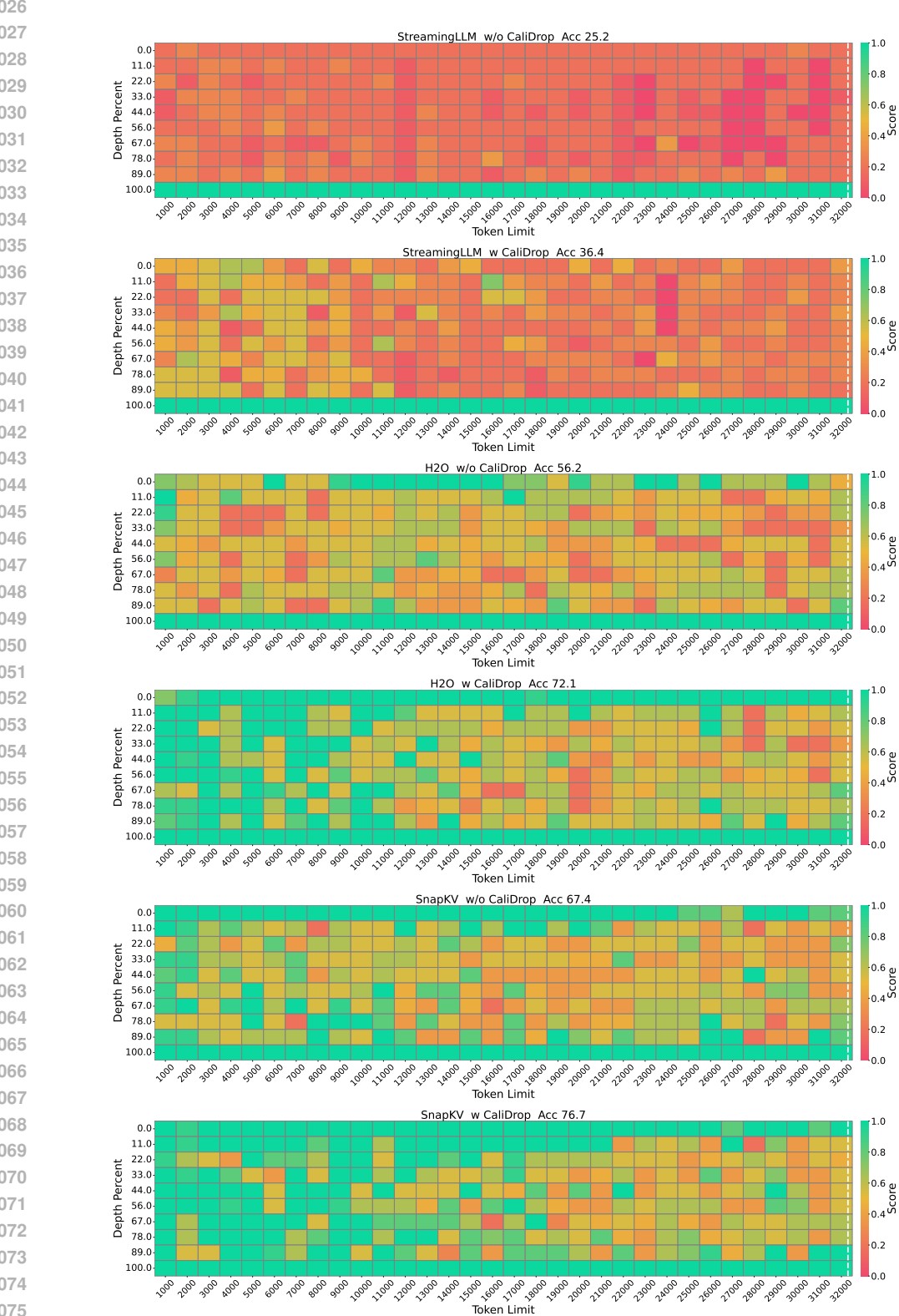

Figure 7: Results of Needle-in-a-Haystack on Mistral-7B-Instruct with 32k context size and 128 KV size. The vertical axis of the figure represents the depth percentage, and the horizontal axis represents the token length.

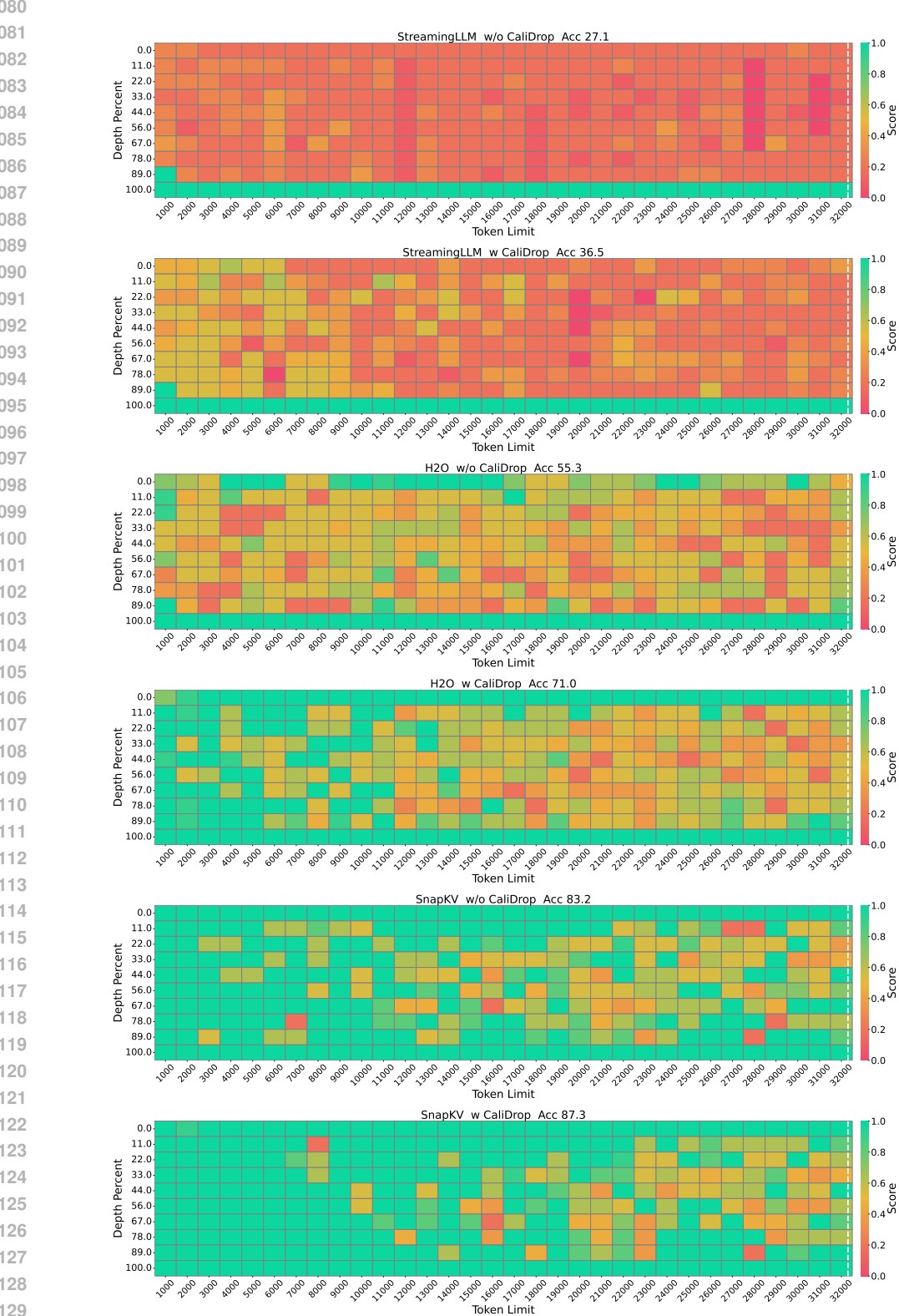

Figure 8: Results of Needle-in-a-Haystack on Mistral-7B-Instruct with 32k context size and 256 KV size. The vertical axis of the figure represents the depth percentage, and the horizontal axis represents the token length.

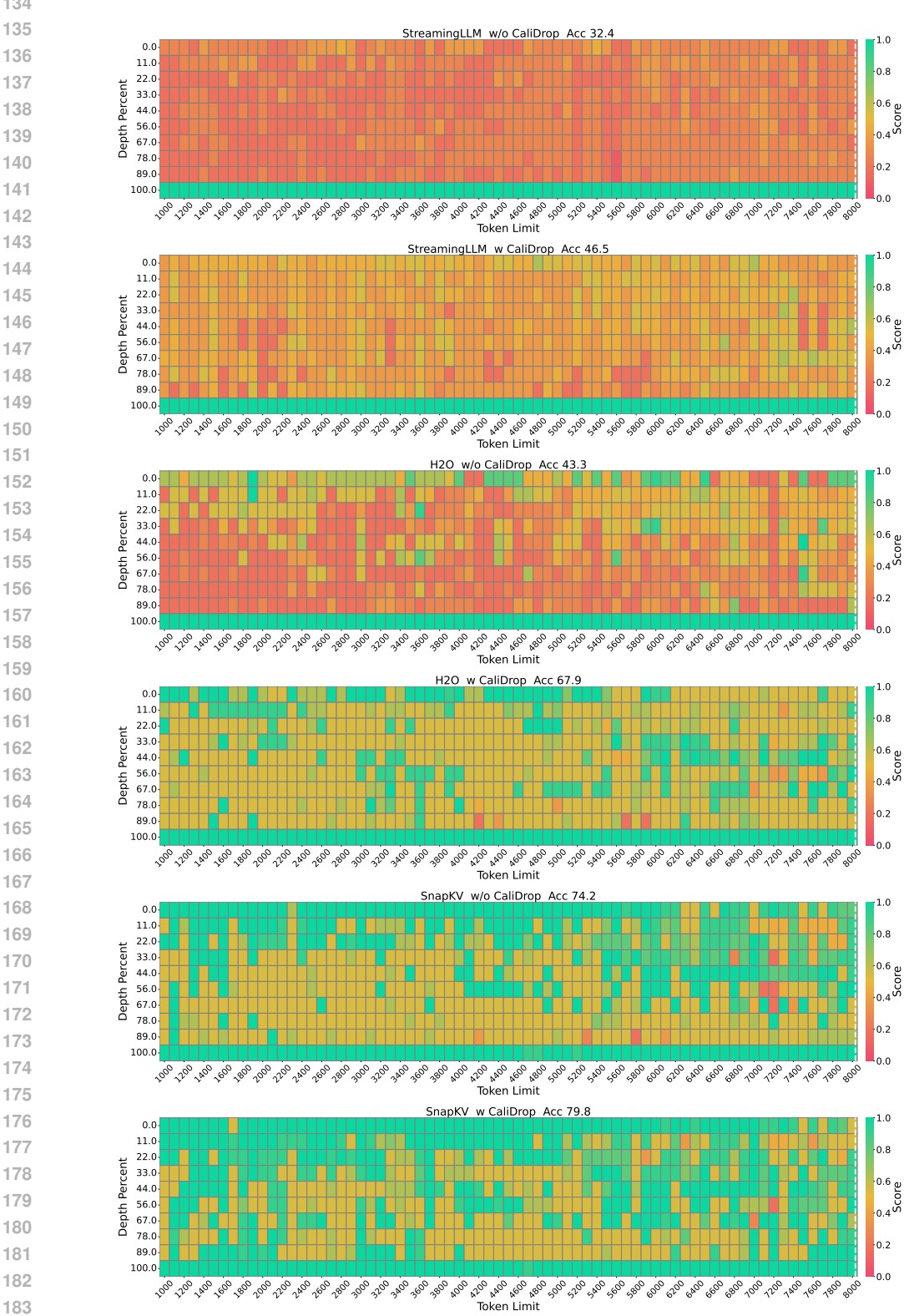

Figure 9: Results of Needle-in-a-Haystack on LLaMA-3-8B-Instruct with 8k context size and 96 KV size. The vertical axis of the figure represents the depth percentage, and the horizontal axis represents the token length.

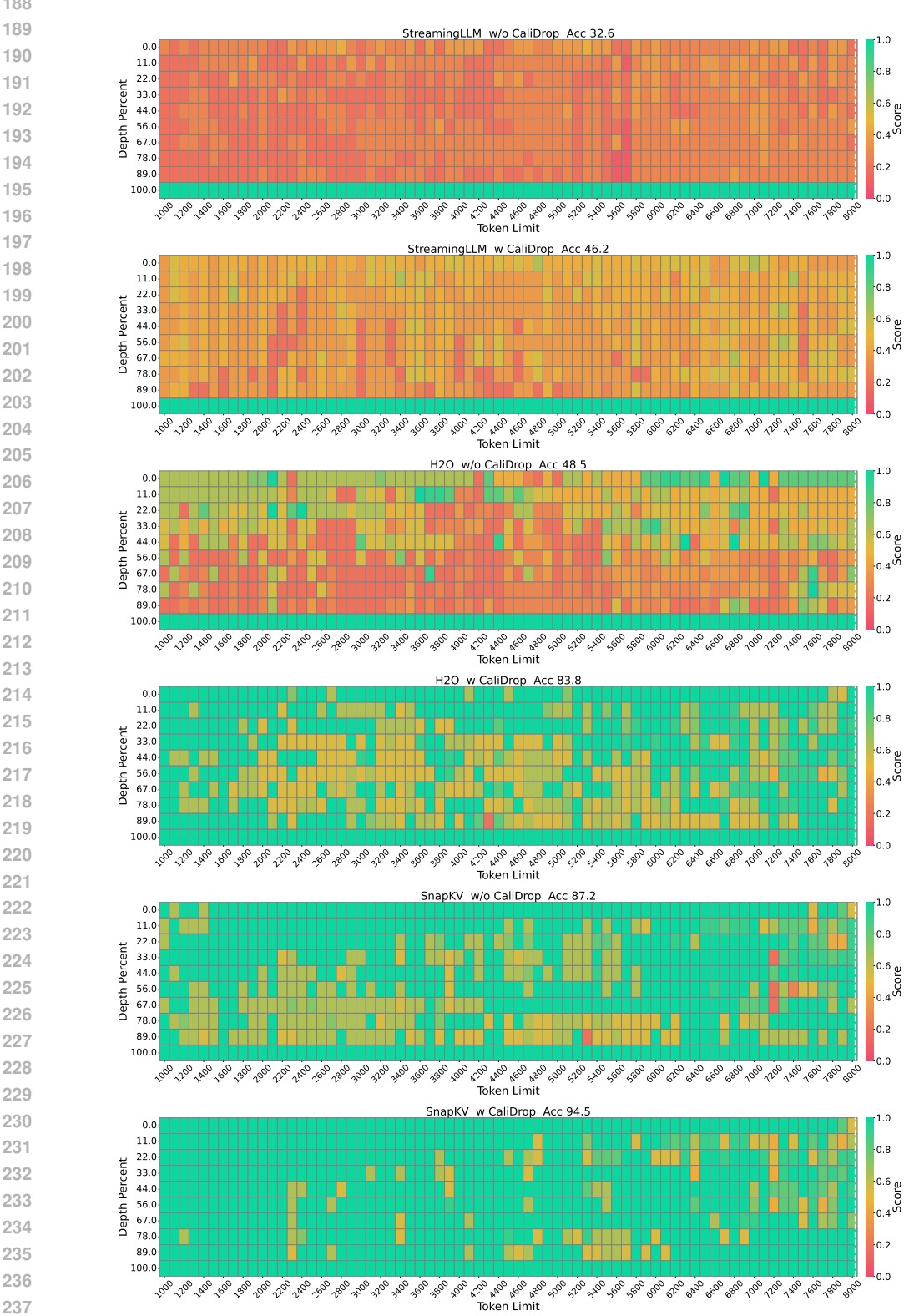

Figure 10: Results of Needle-in-a-Haystack on LLaMA-3-8B-Instruct with 8k context size and 128 KV size. The vertical axis of the figure represents the depth percentage, and the horizontal axis represents the token length.

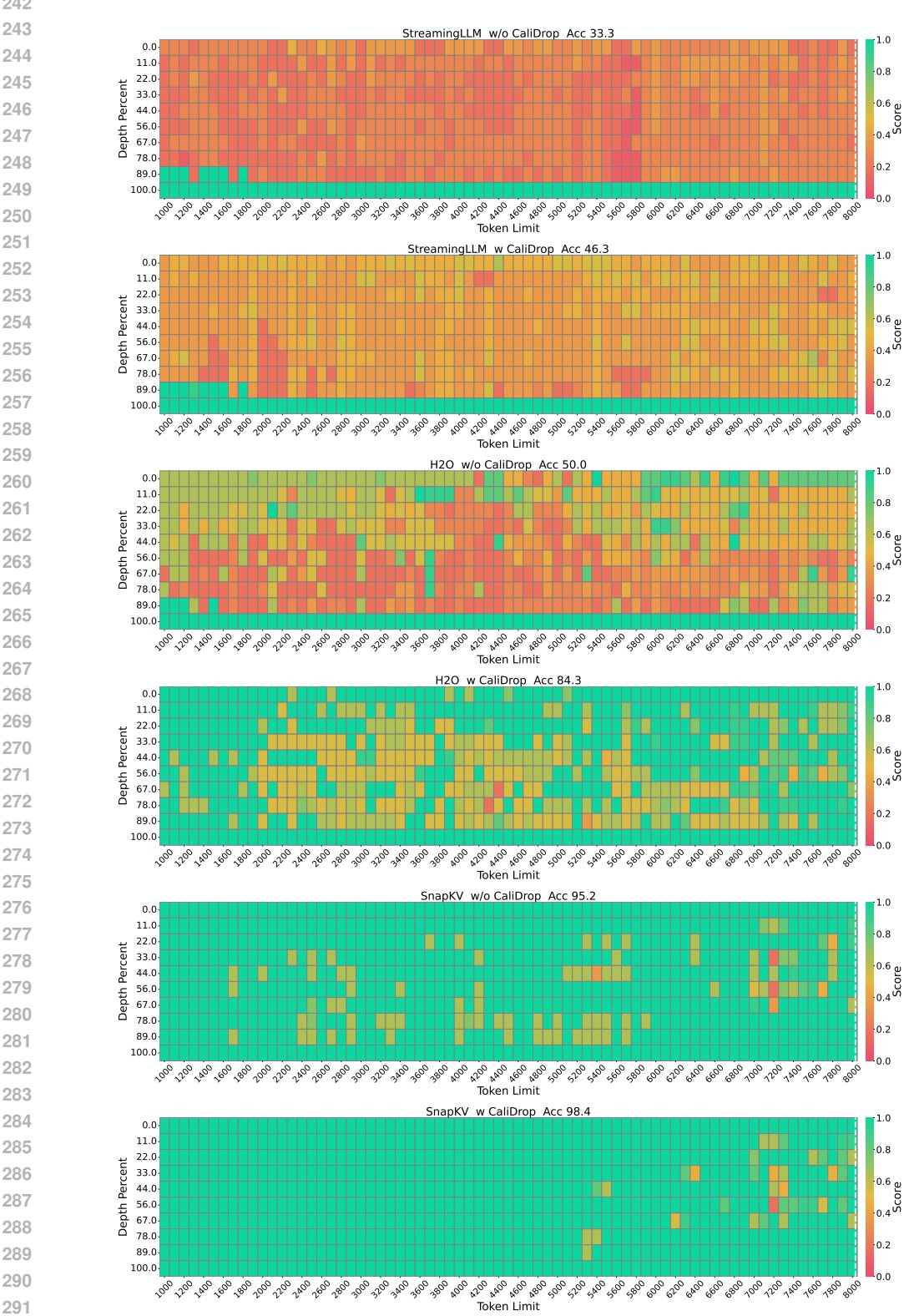

Figure 11: Results of Needle-in-a-Haystack on LLaMA-3-8B-Instruct with 8k context size and 256 KV size. The vertical axis of the figure represents the depth percentage, and the horizontal axis represents the token length.

