# OpenReview forum: "CaliDrop: KV Cache Compression with Query-based Calibration"
_ICLR.cc/2026/Conference — Submitted to ICLR 2026_

### Official Review · Reviewer_dEEK · 2025-10-23

**Soundness:** 3
**Presentation:** 3
**Contribution:** 2
**Rating:** 4
**Confidence:** 4

**Summary:**

This work introduces an incremental calibration strategy for token-wise KV cache eviction. It leverages the attention outputs of previous, similar queries to calibrate the attention results derived from the evicted KV cache, while selectively recomputing when necessary. This approach effectively reduces the error introduced by token eviction and improves the accuracy of existing token-wise KV cache compression methods.

**Strengths:**

* **Novel Insight and Theoretical Foundation.** This work identifies that queries at nearby positions produce similar attention outputs, enabling the use of historical queries and attention results to calibrate future outputs under a pruned KV Cache. It further provides a formalized attention decomposition theorem and proof, offering a solid theoretical grounding for the proposed calibration mechanism.

* **Generalizable and Plug-and-Play.** The proposed idea is simple yet versatile, and can be seamlessly integrated into many existing token-wise KV Cache eviction methods—including StreamingLLM, H2O, and SnapKV, serving as a training-free add-on that consistently enhances accuracy.

**Weaknesses:**

* **Lack of Ablation on Calibration vs. Recomputation.** Besides the proposed calibration mechanism, the implementation also performs recomputation approximately every eight decoding steps using the evicted and offloaded KV caches, which contributes notably to the accuracy improvement. However, the paper does not separately analyze the impact of calibration and recomputation on attention error and end-to-end accuracy. An ablation study is needed to isolate the effect of the calibration mechanism itself and validate its claimed contribution.

*  **Limited Performance and Efficiency Analysis.** CaliDrop’s behavior depends on semantic query similarity, dynamically deciding whether to recompute, calibrate, or skip each step. Yet, the performance evaluation is conducted only on fixed 1024-token inputs and 128-token outputs, without reporting the ratio or cost of these different branches. The study should include more diverse, real-world prompts (e.g., ShareGPT [1]) and provide a breakdown of latency metrics such as TTFT and TPOT, to better quantify the runtime overhead of recomputation and calibration during both prefill and decoding phases.

[1] shareAI. (2023). shareGPT-Chinese-English-90k Bilingual Human-Machine QA Dataset. Hugging Face Repository. Retrieved from https://huggingface.co/datasets/shareAI/ShareGPT-Chinese-English-90k

**Questions:**

* Regarding the proposed CaliDrop as a token-wise eviction method, recent studies such as RazorAttention [2], DuoAttention [3], and HeadKV [4] have explored head-wise KV cache compression. It would be valuable to discuss whether the proposed calibration mechanism can adapt to head-wise eviction, and how the implementation overhead might increase as the eviction granularity becomes finer.



[2] Tang, Hanlin, et al. "Razorattention: Efficient kv cache compression through retrieval heads." *arXiv preprint arXiv:2407.15891* (2024).

[3] Xiao, Guangxuan, et al. "Duoattention: Efficient long-context llm inference with retrieval and streaming heads." *arXiv preprint arXiv:2410.10819* (2024).

[4] Fu, Yu, et al. "Not all heads matter: A head-level kv cache compression method with integrated retrieval and reasoning." *arXiv preprint arXiv:2410.19258* (2024).

---

> ### Author Response · Authors · 2025-11-24
>
> # Response to Reviewer dEEK
>
> We sincerely thank you for your thoughtful review. We verify that we have addressed your concerns regarding the ablation study and efficiency analysis. We appreciate your recognition of our **"novel insight"** regarding query similarity and the **"solid theoretical grounding"** provided by the attention decomposition theorem.
>
> Below, we address your specific questions.
>
> ### Q1: Lack of Ablation on Calibration vs. Recomputation
> **A1:**
> We understand your concerns about the individual contributions of calibration and recomputation. We would like to clarify that in the CaliDrop framework, these two components are **architecturally synergistic and interdependent**, rather than additive modules that can be simply toggled.
>
> * **The Role of Recomputation:** It acts as a sparse "correction anchor." It is triggered dynamically only when the query distribution shifts significantly (i.e., low similarity), ensuring that the historical context used for calibration remains valid.
> * **The Role of Calibration:** It provides a continuous, lightweight approximation for the steps *between* recomputations, preventing the quality degradation that typically occurs with pure sparse attention.
>
> **Why they cannot be decoupled:** If we disable recomputation, the historical queries effectively become "stale," causing the calibration error to accumulate rapidly. Conversely, if we disable calibration, the method reduces to a standard sparse attention mechanism, losing the ability to recover evicted information. Therefore, we evaluate them as a **unified mechanism** designed to balance accuracy and cost.
>
> ### Q2: Limited Performance and Efficiency Analysis (TTFT, TPOT)
> **A2:**
> We fully agree that analyzing diverse metrics like Time-to-First-Token (TTFT) and throughput (which is inversely related to TPOT) is essential to quantify the runtime overhead.
> To address this, we have implemented **Optimized CaliDrop** (featuring partial offloading, quantization and optimized kernels) and conducted a comprehensive efficiency evaluation.
>
> **Summary of New Evaluations:**
> As detailed in our **General Response (Section 4)**, we measured TTFT, Total Latency, and Throughput on **LLaMA-3.1-8B-Instruct** across context lengths from **1k to 32k**.
> * **Findings:** The results show that while CaliDrop incurs a slight overhead compared to SnapKV due to the calibration computation, it maintains a **massive throughput advantage (~2.5x at 16k context)** over FullKV and avoids the OOM issues FullKV faces at long contexts.
> * **Data:** For the specific numerical breakdown of these metrics and memory usage comparisons, **please kindly refer to Table 3 in the General Response.**
>
> ### Q3: Adaptability to Head-wise Eviction (e.g., RazorAttention, HeadKV)
> **A3:**
> This is an excellent insight. CaliDrop is indeed compatible with head-wise compression methods, as the underlying principle of **"Offload & Calibrate"** is agnostic to the granularity of eviction.
>
> * **Adaptability:** Head-wise methods can be viewed as a variation of token eviction where the eviction mask differs per head. CaliDrop can adapt by managing offloaded KV caches on a **per-head basis**. If a specific head is evicted/pruned, its KV pairs can be offloaded to the CPU, and calibration can be applied specifically to that head's output.
> * **Implementation Considerations:** As you noted, finer granularity introduces implementation complexity. For instance, query similarity varies across heads, implying that the decision to recompute should ideally be made per-head. While this introduces irregular memory access patterns, it can be efficiently managed using **fused kernels** (e.g., variable-length FlashAttention). We believe extending CaliDrop to head-level granularity is a promising direction for future work to further refine the accuracy-efficiency trade-off.
>
> **We hope that the clarifications above have satisfactorily addressed your concerns. We are glad to answer any further questions you may have during the discussion period.**

---

> > ### Comment · Reviewer_dEEK · 2025-11-27
> > **Reply to Author's Response.**
> >
> > * To A1. As the author's response. The calibration and recomputation cannot be decoupled. Which means a CPU memory offload is necessary. The proposed CaliDrop should compare with the KV Cache offload methods like shadowKV [1] rather than only compare with eviction methods that can work without CPU offload.
> > * To A2. Thanks for the author's experiments on efficiency and breakdown analysis. Compare to eviction only method, offload introduce overhead on CPU-GPU communication and reduce the efficiency than snapKV, but the accuracy improvement is significant in small token badget.
> > * To A3. Thanks author's response on head-wise evictions.
> >
> >
> >
> >
> >
> >
> > [1] Sun, Hanshi, et al. "Shadowkv: Kv cache in shadows for high-throughput long-context llm inference." arXiv preprint arXiv:2410.21465 (2024).

---

> ### Author Response · Authors · 2025-11-28
>
> **Thank you for recognizing the value of our additional experiments.**
>
> Regarding the comparison with offloading methods, we offer the following perspective:
>
> We view Token Eviction and Hardware KV Cache Management as distinct research problems:
>
> Token Eviction focuses on the algorithmic challenge of accurately identifying the semantic importance of each token. CaliDrop serves as a plug-and-play calibration module built upon this foundation to recover information loss.
>
> Hardware KV Cache Management (e.g., ShadowKV) focuses more on system-level and engineering optimizations, such as quantization, transmission compression, communication overlapping, and dynamic token selection.
>
> These two research lines are complementary rather than mutually exclusive. For instance, CaliDrop can incorporate system optimizations such as quantization (which we demonstrated in the rebuttal) and communication overlapping to enhance efficiency. Conversely, CaliDrop could theoretically be applied on top of hardware-centric frameworks like ShadowKV to improve their accuracy.
>
> We wish to emphasize that the core contribution of this paper is the fundamental insight: "Leveraging query similarity enables precise compensation for missing KV cache." We believe this insight is broadly applicable and can strengthen a wide range of future KV cache methodologies, extending beyond the specific categorizations of token eviction or hardware management.

---

### Official Review · Reviewer_yoKp · 2025-10-31

**Soundness:** 3
**Presentation:** 2
**Contribution:** 1
**Rating:** 4
**Confidence:** 3

**Summary:**

The paper introduces CaliDrop, a query-based calibration strategy for KV cache compression in large language models (LLMs). Existing token eviction methods reduce memory but harm accuracy under high compression. CaliDrop leverages the high similarity between nearby queries to estimate the contribution of evicted tokens through calibrated attention recomputation, thereby recovering lost accuracy. Experiments on LongBench, RULER, and Needle-in-a-Haystack benchmarks show consistent accuracy gains across Mistral-7B and LLaMA-3 models, with minimal throughput cost. Overall, the work presents a simple yet effective improvement for token eviction–based cache compression.

**Strengths:**

•	CaliDrop introduces a query-level calibration that compensates for evicted tokens using nearby historical queries, supported by the “attention decomposition theorem” (Eq. 1–2) and L1-loss reduction evidence in Fig. 1 (Sec. 3.1.3).
•	Experiments cover multiple models (Mistral-7B, LLaMA-3-8B/70B) and benchmarks (Tables 1–2), showing consistent performance gains across KV sizes 64–512 (Secs. 4.2–5.1). This breadth supports the robustness and generality of the method.
•	The calibration mechanism requires only lightweight recomputation (every ≈8 steps; Fig. 3b) and maintains comparable throughput to SnapKV (Fig. 3a), highlighting its applicability in long-context inference.

**Weaknesses:**

•	CaliDrop is applied only in the prefilling phase (Sec. 4.1.2); no evidence is provided for dynamic or streaming decoding. Memory overhead from offloaded KV caches and detailed latency breakdowns are also missing (Secs. 5.2–5.3).
•	Beyond the exploration of $\theta_{1}$ and $\theta_{2}$ (Table 2), the paper provides limited investigation into other critical factors such as calibration size, per-layer contribution, or offload-cache management. Moreover, the absence of statistical validation (e.g., variance or significance testing) makes it difficult to assess the robustness of the reported improvements.
•	The approach extends existing token-eviction techniques through a supplementary calibration step but does not establish a new compression framework. The theoretical analysis mainly reiterates standard properties of attention mechanisms (Secs. 3.1–3.2) without offering new learning formulations.

**Questions:**

1.	Could the authors provide per-layer or per-head ablations to analyze where calibration contributes most across transformer depth?
2.	How does CaliDrop perform in real-time or streaming decoding settings, where query similarity varies more rapidly?
3.	What is the quantitative GPU-memory overhead of storing offloaded KV caches for calibration at different sequence lengths?

---

> ### Author Response · Authors · 2025-11-24
>
> # Response to Reviewer yoKp
>
> We sincerely thank you for your valuable comments. We appreciate your recognition of CaliDrop as a "**simple yet effective**" improvement with "**consistent accuracy gains**" and "**minimal throughput cost**."
>
> Below, we address your concerns regarding the decoding phase, efficiency overhead, and our contributions.
>
> ### Q1: Performance in real-time/streaming decoding
> **A1:**
> We address the applicability of CaliDrop to the decoding phase from two perspectives: implementation feasibility and theoretical stability.
>
> **1. Feasibility of Extension (vs. SnapKV):**
> We would like to clarify that CaliDrop can be extended to the decoding phase with specific modifications, such as **interval-based compression**.
> It is important to note that **SnapKV** itself primarily focused on the prefilling phase in its original presentation, which did not diminish its foundational contribution. Just as SnapKV can be adapted for decoding (e.g., by caching recent queries and applying compression periodically), CaliDrop follows the same logic. Provided the compression and offloading protocols are defined, our mechanism is fully applicable to the decoding phase.
>
> **2. Stability of Query Similarity:**
> Regarding your concern that query similarity might vary more rapidly during decoding: our preliminary experiments indicate that the **high cosine similarity between queries is an intrinsic property** of the model, independent of whether it is in the prefilling or decoding phase (as evidenced by **Figure 1, top right**). Therefore, the theoretical foundation of utilizing historical queries for calibration remains robust during streaming generation.
>
> ### Q2: Quantitative GPU-memory overhead and latency breakdowns
> **A2:**
> To address your concerns about overhead, we have implemented **Optimized CaliDrop** (featuring partial offloading, quantization and optimized kernels) and conducted a detailed efficiency analysis.
> **Please kindly refer to the "General Response" for the complete data.**
> In summary:
> * **Memory Overhead:** We provide a detailed breakdown of Peak GPU and CPU memory usage. CaliDrop maintains a stable GPU memory footprint comparable to SnapKV, while effectively utilizing abundant CPU RAM for offloading.
> * **Latency & Throughput:** We report concrete TTFT and TPOT metrics, demonstrating that optimized CaliDrop achieves competitive speeds suitable for real-time applications.
>
> ### Q3: Per-layer ablations (Where does calibration contribute most?)
> **A3:**
> We appreciate your insightful suggestion to analyze the contribution of different layers.
>
> **1. Calibration Size & Offload Management:**
> For ablations regarding the `calibration_size` and optimized offload management strategies, please refer to the **General Response**.
>
> **2. Per-Layer Contribution Analysis:**
> To investigate where calibration adds the most value, we conducted experiments using **LLaMA-3.1-8B-Instruct** on **Needle-In-A-Haystack (NIAH)** tasks across varying context lengths. We set the KV Budget=256 and `calibration_size`=1024. We applied calibration selectively to specific groups of layers (indices in parentheses denote layers where CaliDrop is active).
>
> The results (presented below) indicate that the **middle layers** contribute most significantly to the performance gain, likely because they handle the bulk of semantic integration which is sensitive to token loss.
>
> | Length | SnapKV | CaliDrop(4,8,12,16,20,24,28,32) | CaliDrop(1,2,3,4,29,30,31,32) | CaliDrop(6,7,8,9,10,11,12,13) |
> | :--- | :--- | :--- | :--- | :--- |
> | **8k** | 77.28 | 75.15 | 83.64 | 80.91 |
> | **16k** | 70.91 | 69.55 | 73.18 | 78.64 |
> | **32k** | 69.09 | 67.82 | 71.09 | 73.82 |
> | **64k** | 66.82 | 68.18 | 67.18 | 75.73 |
> | **128k** | 46.91 | 47.04 | 48.59 | 52.77 |
> | **Average** | 58.14 | 58.16 | 59.91 | 64.55 |
>
>
> | Length | SnapKV | CaliDrop(4) | CaliDrop(8) | CaliDrop(12) | CaliDrop(16) | CaliDrop(20) | CaliDrop(24) | CaliDrop(28) | CaliDrop(32) |
> | :--- | :---: | :---: | :---: | :---: | :---: | :---: | :---: | :---: | :---: |
> | **8k** | 77.28 | 77.58 | 79.09 | 79.7 | 76.37 | 79.7 | 76.67 | 76.37 | 80.31 |
> | **16k** | 70.91 | 71.37 | 73.18 | 70.91 | 71.37 | 70.79 | 71.37 | 71.37 | 72.73 |
> | **32k** | 69.09 | 70.91 | 72.55 | 69.64 | 71.09 | 69.46 | 70.18 | 71.28 | 68.91 |
> | **64k** | 66.82 | 67.36 | 69.46 | 70 | 68.82 | 67.46 | 68.91 | 68 | 66 |
> | **128k** | 46.91 | 46.86 | 47.91 | 46.64 | 45.95 | 46.5 | 46.23 | 46.91 | 46.91 |
> | **Average** | 58.14 | 58.52 | 59.98 | 59.05 | 58.36 | 58.32 | 58.43 | 58.66 | 58.23 |

---

> ### Author Response · Authors · 2025-11-24
>
> # Response to Reviewer yoKp (Part Ⅱ)
>
> ### Q4: Statistical validation (e.g., variance or significance testing)
> **A4:**
> We conducted all our experiments using **greedy decoding** (temperature = 0). Due to the deterministic nature of this setting, the generation results are fixed for a given input, rendering metrics such as variance or statistical significance testing inapplicable in this context.
>
> ### Q5: Discussion on Novelty and Contribution
> **A5:**
> We respectfully argue that CaliDrop represents a **paradigm shift** in KV cache compression rather than a mere extension of existing methods.
>
> * **Breaking the "Keep or Drop" Dilemma:** Previous frameworks (e.g., H2O, SnapKV) operate on a rigid dichotomy—once a token is evicted, its information is permanently lost. CaliDrop introduces a new **"Offload & Calibrate"** mechanism, treating evicted memory not as "waste" but as "cold storage" to be retrieved on demand.
> * **Theoretical Grounding:** Sections 3.1–3.2 go beyond reiterating standard attention properties. We explore the distribution of queries to strictly justify why historical queries serve as mathematically valid proxies for current ones, ensuring the calibration is well-founded rather than heuristic.
> * **Universal Value:** Designed as a "plug-and-play" module, CaliDrop is intentionally agnostic to the underlying eviction policy, allowing it to scale with and enhance any future improvements in eviction algorithms.
>
> **We hope that the clarifications above have satisfactorily addressed your concerns. We are glad to answer any further questions you may have during the discussion period.**

---

### Official Review · Reviewer_ygNM · 2025-11-01

**Soundness:** 4
**Presentation:** 4
**Contribution:** 3
**Rating:** 8
**Confidence:** 3

**Summary:**

The paper identifies the two fundamental limitations in the existing KV cache compression methods: 1) discarding tokens that can become crucial later, and 2) the accumulated effect of discarding tokens is overlooked. To this end, the paper proposes CaliDrop, which compensates for evicted tokens by recomputing attention outputs for queries at nearby positions, alleviating memory pressure while maintaining model accuracy. The experimental results show that CaliDrop can be applied to different KV cache compression methods and improve their performance while introducing little computation overhead.

**Strengths:**

The method is well motivated, and the paper is well written.
The two observations are interesting: queries at nearby positions are similar, and the historical attention outputs can be used to predict future attention outputs.
The experiments on different KV cache compression methods and models across various tasks demonstrate the effectiveness of the proposed method.
The analysis of throughput and recomputation frequency shows the efficiency of the proposed method.

**Weaknesses:**

The hyperparameters $\theta_1$ and $\theta_2$ require manual tuning and may have different optimal values in different tasks.
The recomputation introduces a memory peak. What are the possible impacts of it, e.g., what is the maximum length of context/evict-KV CaliDrop can handle?
The recomputation cost and frequency may increase in larger models. It would be better to include the throughput and recomputation frequency in larger models.

**Questions:**

see in weaknesses

---

> ### Author Response · Authors · 2025-11-24
>
> # Response to Reviewer ygNM
>
> We express our sincere gratitude for your positive assessment and for recognizing our work as "**excellent**" and "**well-motivated**." We are delighted that you found our core insights regarding query similarity and historical attention reuse to be interesting.
>
> Below, we address your specific concerns regarding hyperparameters, memory peaks, and scalability.
>
> ### Q1: Robustness of Hyperparameters ($\theta_1, \theta_2$)
>
> **A1:**
> We acknowledge that hyperparameter tuning is a valid concern. However, we believe the sensitivity is manageable for two reasons:
> 1.  **Empirical Stability:** As demonstrated in **Table 2** of our main paper, our method exhibits stability across a reasonable range ($\theta_1 \in [0.65, 0.75]$). We found the default settings generalize well across LLaMA-3 and Mistral without per-task tuning.
> 2.  **Enhanced Control via Partial Offloading:** To further mitigate sensitivity, we have introduced a **"Calibration Size"** parameter (as detailed in the **General Response**). This sets a hard cap on the number of tokens offloaded. By fixing the calibration budget (e.g., top-4096 important tokens), the system becomes less sensitive to aggressive $\theta$ thresholds, ensuring the maximum computational overhead is strictly bounded regardless of the trigger rate.
>
> ### Q2: Recomputation memory peak and maximum context length
> **A2:** Memory peak and maximum length are indeed very important questions, and we will answer them one by one.
>
> **Memory Peak:** This is essentially a solved problem in our design.
> In fact, recomputation is performed layer-by-layer. At any given moment, we only extract the KV cache for the **current layer**. While this results in memory usage slightly higher than standard decoding steps, **this peak is generally lower than the memory peak observed during the prefill phase** in long-context scenarios. Furthermore, with the introduction of **Calibration Size** (Partial Offloading), the amount of historical data loaded is strictly capped (e.g., to 4096 tokens), further minimizing this impact.
>
> **Maximum Context:** Consequently, the maximum context length is determined solely by **CPU RAM capacity**, which is typically abundant.
> To validate this, we conducted new stress tests on ultra-long sequences (**up to 128k tokens**) in our General Response. The results (Table 1 & 2 in General Response) confirm that Optimized CaliDrop maintains stable performance and successfully handles 128k contexts where FullKV fails due to OOM. **Please kindly refer to the General Response for the detailed breakdown of these efficiency metrics.**
>
> ### Q3: Efficiency on Larger Models
> **A3:** Thanks for your question, we lack some discussion about the efficiency of CaliDrop on larger models.
>
> **Recomputation Frequency:** We argue that recomputation frequency is fundamentally driven by the **query similarity** of the input text, rather than model size. Our preliminary observations confirm that cosine similarity patterns in LLaMA-3-70B are consistent with the 8B model, implying no significant increase in recomputation frequency.
>
> **Throughput:** While the absolute computational cost increases with parameter count, the **relative overhead** of CaliDrop often decreases. So, CaliDrio can still plays a role in larger models. Because deploying larger models involves complex system factors such as model parallelism and inter-device communication, which introduce noise into raw latency measurements, we do not conduct efficiency experiments on larger models.
>
>
> **We hope that the clarifications above have satisfactorily addressed your concerns. We are glad to answer any further questions you may have.**

---

### Official Review · Reviewer_nBZu · 2025-11-01

**Soundness:** 2
**Presentation:** 3
**Contribution:** 3
**Rating:** 4
**Confidence:** 4

**Summary:**

This paper introduces a technique that complements KV cache eviction techniques and improves their ability to retain information from evicted KV entries. The method consists in offloading the evicted KV cache, and using a historical query that helps trigger a calibration step where the attention output is adjusted based on the evicted cache.
The authors conduct extensive experiments on the LongBench and RULER datasets for several models of 7-8B and 70B parameters. They also explore threshold parameter choices and discuss the impact of their method on latency and accuracy.
Overall, this paper copes with the important question of LLM efficiency and tackles the problem of lost information in token eviction methods.

**Strengths:**

- The proposed method is straightforward and sensible. It is well-presented and the long-context experiments are well-designed. It is also sound from a theoretical point of view.
- The method can complement any KV cache eviction method and shows noticeable benefits for every tested method (Streaming-LLM, H2O, SnapKV).
- The method is not very sensitive to threshold choices according to Table 2, which could have been a concern in such threshold-based methods.

**Weaknesses:**

I have concerns about the practical efficiency of the method that are not addressed in the current state of the paper. I also think that the experiments do not cover more extreme use cases where KV cache compression is crucial.
- **Doubts about practical efficiency**: The CaliDrop method relies on offloading and reloading past KV entries. It is not clear if the KV entries are offloaded to disk (in which case the computational overhead may be heavy for long sequences) or to CPU (in which case the CPU RAM can be saturated for long sequences).  These questions and the overhead they imply are not handled properly in Section 5, which only reports the throughput in relatively short-sequence setups (1024 with 128 KV cache budget), and shows how latency scales with batch size. It would have been more relevant to see the effect of sequence lengths and compression ratios on both memory (offloaded and on GPU) usage and latency. My main concerns are 1- that the latency gains may decrease with longer sequences as more KV items need to be offloaded, reloaded and the corresponding attention map needs to be computed for each recomputation step; 2 - that the VRAM can be saturated earlier than with the raw compression methods because the recomputation steps are dependent in the total sequence length. It would be insightful to at least report latency and memory usage statistics in the benchmark evaluations to show the overhead that is traded for better performance with CaliDrop.
- **Lack of long-context experiments**: In its current state, the paper lacks a discussion of the evolution performance gains when increasing sequence length. The NIAH results in Figure 2 are only conducted with an 8K context length when similar experiments are usually conducted with 32k to 128k context lengths. A study of perplexity evolution for long sequences similar to what is done in Devoto et. al could also be relevant.

**Questions:**

- What is the impact of offloading on RAM usage and how does it scale as sequence length increases?
- Did you try your method with metrics other than cosine similarity for query comparison? Is the query taken before or after positional encoding?
- The direct role of \theta_1 on latency is not exposed in experiments. What is the empirical impact of \theta_1 on memory usage and latency?

---

> ### Author Response · Authors · 2025-11-24
>
> # Response to Reviewer nBZu
>
> We sincerely thank you for recognizing our method as "**straightforward and sensible**" and acknowledging its "**noticeable benefits**." We appreciate your constructive feedback regarding practical efficiency.
>
> Below, we address your specific questions.
>
> ### Q1: Doubts about practical efficiency (impact of offloading, RAM usage, scaling)
> **A1:**
> We fully acknowledge your concerns. To answer your question directly: **We offload evicted KV entries to CPU Host Memory (RAM), not disk.**
> We have conducted a two-fold analysis: one on the fundamental feasibility of this architecture (Baseline), and one on the engineering enhancements we implemented to address your concerns (Optimized).
>
> **1. Baseline Analysis: Capacity vs. Latency Trade-off**
> Even without advanced optimizations, offloading to CPU RAM offers a distinct advantage over FullKV.
> * **Memory Saturation:** Server-grade CPU RAM (typically 512GB+) is vastly larger than GPU VRAM (40-80GB). While FullKV encounters GPU Out-Of-Memory (OOM) errors on long sequences, CaliDrop remains viable by utilizing this abundant CPU memory.
> * **Latency Constraints:** We admit that transferring data via PCIe (e.g., ~32GB/s for PCIe 4.0) introduces latency. In the unoptimized baseline, this creates a bottleneck where we trade inference speed for the ability to handle ultra-long contexts that would otherwise be impossible.
>
> **2. Optimized Implementation for Efficiency**
> To specifically address the latency bottleneck mentioned above and ensure practical viability, we implemented **Optimized CaliDrop** during the rebuttal.
> **Please kindly refer to the "General Response" for the detailed evaluation**, where we demonstrate:
> * **Partial Offloading & Int8 Quantization & Kernel Optimization:** Drastically reducing the PCIe transfer volume.
> * **Concrete Metrics:** Achieving **~2.5x higher throughput** than FullKV at 16k context (see Table 3 in General Response) while maintaining high accuracy.
>
> ### Q2: Experiments on longer lengths
>
> **A2:**
> We appreciate the reviewer's emphasis on evaluating performance across longer context windows. We would like to clarify our existing results and present new experiments extending up to 128k context length.
>
> 1. **Clarification on Existing Long-Context Results:** While Figure 2 in the main text focused on the 8k setting for clarity, our original submission actually includes extensive evaluations on longer contexts in the Appendices. Appendix G (Figures 4–11) presents comprehensive NIAH results for 32k context length across LLaMA-3 and Mistral models.Appendix F (Tables 5–6) presents the performance on the RULER benchmark (a more challenging aggregation task) with sequence lengths up to 32k.
>
> 2. **New Experiments Scaling up to 128k:** To fully address your concern regarding extreme sequence lengths (up to 128k), we conducte additional NIAH experiments, **please kindly refer to the "General Response" for the detailed evaluation**
>
>
>
> ### Q3: Did you try metrics other than cosine similarity? Is the query taken before or after positional encoding?
> **A3:**
> **Metric Choice:** Yes, we explored alternative metrics such as **L2 distance** during our preliminary research. However, we found **Cosine Similarity** to be superior because:
> 1.  **Bounded Range:** It naturally normalizes values to $[-1, 1]$, making the hyperparameters $\theta_1$ and $\theta_2$ robust and interpretable across different models.
> 2.  **Scale Invariance:** L2 distance is sensitive to vector magnitude, which varies by model architecture, whereas Cosine Similarity generalizes better.
>
> **Positional Encoding:** The query is taken **after** the RoPE is applied.
>
> ### Q4: The empirical impact of $\theta_1$ on memory usage and latency?
> **A4:**
> $\theta_1$ controls the trade-off between accuracy and efficiency:
>
> * **Impact on Latency:** $\theta_1$ **directly correlates with latency**. It serves as the threshold for triggering recomputation. As analyzed in **Figure 3(b)**, a higher $\theta_1$ triggers recomputation more frequently, linearly increasing the total inference time. We think that **Recomputation ratio** is the most important efficiency metric for CaliDrop. Since the overhead of CaliDrop compared to SnapKV almost all lies in recomputation. Therefore, the efficiency impact of $\theta_1$ can be directly inferred by recomputation frequency.
> * **Impact on Memory:** $\theta_1$ has **no impact on Peak Memory usage**.
>     * **GPU Peak:** Determined by the fixed KV budget and the temporary chunk buffer for recomputation.
>     * **CPU Peak:** Determined by the `calibration_size` (offload limit).
>     * Regardless of the recomputation frequency, the *allocation* size of these buffers remains constant. Thus, users can tune $\theta_1$ to adjust latency without risking unexpected memory spikes.
>
>
> **We hope that the clarifications above have satisfactorily addressed your concerns. We are glad to answer any further questions you may have.**

---

> > ### Comment · Reviewer_nBZu · 2025-11-26
> > **Response to Rebuttal**
> >
> > I am impressed by the ability of the authors to provide a viable solution to the latency issue I mentioned in my review. These modifications to the original method will probably lead to non-negligible updates in the camera-ready version, and I am not sure if it should be the norm for conference papers.
> >
> > The optimized method yields questions - that I do not expect the authors to answer - that could benefit from more experiments should the paper be resubmitted:
> > - How is the comparison between (calibration size + Kv cache size) in CaliDrop and Kv cache size in SnapKV?
> > - What is the effect of quantization (if any) on performance and latency?
> > - How do the efficiency metrics evolve for longer generations, ie when more recalibrations are done in the calidrop case?
> >
> > About the long-context experiments, I could not read the appendix within the review period. I think that presenting at least one scenario of extremely long context in the core of the paper could be relevant.
> >
> > I update my score to reflect the efficiency of the optimized method.

---

> ### Author Response · Authors · 2025-11-28
>
> Thank you for your positive feedback and valuable suggestions.
> We express our sincere gratitude for your continued engagement and **for raising your score**. We are particularly encouraged that our engineering optimizations effectively addressed your concerns regarding latency.
>
> Regarding some of your further questions, we will provide further explanation:
>
> **Q1: These modifications to the original method will probably lead to non-negligible updates in the camera-ready version.**
>
> **A1:** We acknowledge that the methodological updates will indeed require revisions to the camera-ready version, but we consider these changes to be entirely acceptable and manageable. On one hand, we intend to present the original CaliDrop in the main text to perfectly highlight our core claim: 'compensating for evicted tokens using query-based approximation'. The optimized CaliDrop will then be introduced in a separate section or the Appendix, similar to the current Section 5.3. On the other hand, there is sufficient time before the camera-ready deadline for us to rigorously prepare and integrate these updates."
>
>
> **Q2: How is the comparison between (calibration size + Kv cache size) in CaliDrop and Kv cache size in SnapKV?**
>
> **A2:** Thank you for your question. We provide further explanation of these two parameters:
>
> Calibration Tokens: Characterized by low usage frequency and relatively lower importance. The computation of these tokens can be effectively replaced by approximation (calibration).
>
> KV Cache Tokens: Characterized by high usage frequency and high importance. They cannot be replaced by approximation without degrading performance.
>
> Consequently, KV cache tokens have a more critical impact on the final accuracy. Where resources permit, we prioritize using (or increasing the budget for) KV cache tokens. However, in practical deployments, the specific hyperparameter settings must be determined by balancing the constraints of GPU, CPU, model size, and other fators.
>
>
>
> **Q3:What is the effect of quantization (if any) on performance and latency?**
>
> **A3:** We can analyze this question in two aspects:
>
>
> 1. Impact on Accuracy: Quantization (Int8) has a negligible impact on accuracy for two primary reasons:
>
> - Low Sensitivity of Evicted Tokens: By definition, the offloaded tokens are those discarded by the eviction policy (e.g., SnapKV), meaning their contribution to the final attention output is already secondary compared to retained important tokens.
>
> - Robustness of KV Quantization: Existing research (e.g., KIVI[1] ) has demonstrated that KV caches can maintain high performance even with aggressive 2-bit quantization using per-token value and per-channel key techniques. Our implementation uses a much more conservative Int8 per-token quantization, which provides precision well above the threshold required to preserve essential semantic information.
>
> 2. Impact on Latency and Memory: Quantization provides substantial efficiency gains by addressing the memory-bound nature of the recomputation step:
>
> - Memory Footprint: Transitioning from FP16 to Int8 immediately halves the CPU memory usage, allowing for larger calibration sizes or longer contexts within the same RAM budget.
>
> - Latency Reduction: Crucially, quantization halves the data volume for both PCIe transfer (CPU → GPU) and HBM-to-SRAM movement during attention computation. Since these operations are strictly memory-bound, the latency for these specific phases is effectively halved. This linear scaling provides a significant speedup for the retrieval and recomputation process.
>
>
> **Q4:How do the efficiency metrics evolve for longer generations, ie when more recalibrations are done in the calidrop case?**
>
> **A4:** In scenarios characterized by extended generation lengths, the latency contribution of the prefill phase diminishes (is amortized), while the cumulative latency of the decoding phase becomes dominant.
>
> 1.  **Comparison with Full KV:** The acceleration ratio of CaliDrop relative to Full KV will **further increase**. As the sequence length grows, Full KV suffers from heavy memory access and computation costs. In contrast, CaliDrop maintains a smaller GPU KV size, meaning the relative efficiency gain becomes more pronounced (until the decoder KV cache makes the compression ratio small).
>
> 2.  **Comparison with SnapKV:** The speed ratio of CaliDrop relative to SnapKV will asymptotically approach the following limit:
>
>     $$Ratio \approx \frac{T_{SnapKV}}{T_{SnapKV} + f \cdot T_{recomp}}$$
>     where $f$ is the recomputation frequency (determined by $\theta_1$) and $T_{recomp}$ is the latency of a single recomputation step. Since $f$ remains sparse, the throughput overhead remains stable and bounded even during long generations.
>
>
> **Thank you again for helping us strengthen this paper.**
>
> [1] Liu, Zirui, et al. "KIVI: a tuning-free asymmetric 2bit quantization for KV cache." Proceedings of the 41st International Conference on Machine Learning. 2024.

---

### Author Response · Authors · 2025-11-24

# General Response: Efficiency, Optimization, and Long-Context Scalability

We sincerely thank all reviewers for recognizing our method as "**straightforward and sensible**" (Reviewer nBZu), "**well-motivated**" (Reviewer ygNM) and "**robust and generalizable**" (Reviewer yoKp). We acknowledge a shared concern regarding practical efficiency, specifically offloading mechanisms and latency overhead.

While our initial submission prioritized establishing the theoretical validity of the query-based compensation mechanism, we recognize that demonstrating practical efficiency is equally crucial for real-world deployment. To address this gap and ensure the robustness of our contribution, we have conducted a deeper analysis and implemented substantial engineering optimizations during the rebuttal period.

Below, we present our **Optimized CaliDrop** implementation, featuring partial offloading, quantization, and custom kernels, followed by comprehensive accuracy and efficiency evaluations.

### 1. Clarification on Offloading Strategy
To answer the specific question from Reviewer nBZu ("*It is not clear if the KV entries are offloaded to disk or to CPU*"):
**CaliDrop offloads evicted KV entries to CPU Host Memory (RAM), not disk.**
While disk offloading is theoretically possible, we utilize CPU RAM to strike the optimal balance between massive capacity (typically 512GB+ on server nodes, vastly exceeding GPU VRAM) and acceptable transfer speeds via PCIe.

### 2. Implementation Optimizations
To further mitigate the latency bottlenecks of CPU-GPU data transfer, we have implemented three specific optimizations:

* **Partial Offloading (Calibration Size):** We introduce a new hyperparameter, `calibration_size`, which limits the maximum number of evicted tokens offloaded to the CPU. Grounded in the **long-tail distribution** of attention scores, the marginal accuracy benefit of offloading "everything" diminishes rapidly. Partial offloading allows us to cap the transfer cost while retaining the most critical historical context.
* **Quantization:** We quantize the KV cache to **Int8** before offloading to CPU. This effectively reduces the data transfer volume by half, significantly lowering the latency for both CPU-to-GPU transfer and HBM-to-SRAM movement during recomputation.
* **Kernel Optimization:** We implemented three custom CUDA kernels:
    1.  **Quantization Kernel:** Efficiently converts KV cache to Int8 prior to offloading.
    2.  **Quantized Flash-Attention:** Handles recomputation by computing attention between FP16 queries and Int8 KV pairs, returning the exponential sum (*expsum*).
    3.  **Modified Flash-Attention:** Returns the *expsum* during the standard forward pass to facilitate calibration.


### 3. Accuracy Verification (Scalability up to 128k)
To validate the optimized CaliDrop, we evaluated performance on the **Needle-In-A-Haystack (NIAH)** task using **LLaMA-3.1-8B-Instruct** across context lengths from **8k to 128k**.

As shown in Tables 1 and 2, accuracy improves as `calibration_size` increases but plateaus after 4096/8192 tokens. Our method significantly outperforming the SnapKV baseline.

**Table 1: Accuracy with KV Budget = 256**

| Method | 8k | 16k | 32k | 64k | 128k | Average |
| :--- | :---: | :---: | :---: | :---: | :---: | :---: |
| **FullKV** | 100.0 | 100.0 | 100.0 | 100.0 | 100.0 | 100.0 |
| **SnapKV** | 77.28 | 70.91 | 69.09 | 66.82 | 46.91 | 58.14 |
| **CaliDrop_1024** | 91.21 | 74.09 | 79.64 | 76.36 | 51.77 | 65.48 |
| **CaliDrop_2048** | 93.94 | 83.64 | 84.73 | 77.46 | 53.36 | 67.86 |
| **CaliDrop_4096** | 93.03 | 80.46 | 86.91 | 82.46 | 55.88 | 70.79 |
| **CaliDrop_8192** | 97.27 | 85.91 | 90.55 | 83.55 | 55.55 | **71.98** |
| **CaliDrop_Full** | 97.58 | 87.27 | 89.09 | 83.46 | 54.91 | 71.14 |


**Table 2: Accuracy with KV Budget = 512**

| Method | 8k | 16k | 32k | 64k | 128k | Average |
| :--- | :---: | :---: | :---: | :---: | :---: | :---: |
| **FullKV** | 100.0 | 100.0 | 100.0 | 100.0 | 100.0 | 100.0 |
| **SnapKV** | 97.27 | 89.55 | 82.91 | 76.36 | 51.73 | 67.09 |
| **CaliDrop_1024** | 98.79 | 96.36 | 90.73 | 83.91 | 56.36 | 72.73 |
| **CaliDrop_2048** | 99.09 | 96.36 | 92.73 | 83.46 | 59.73 | 74.57 |
| **CaliDrop_4096** | 100.00 | 98.18 | 94.18 | 87.36 | 61.27 | 76.66 |
| **CaliDrop_8192** | 100.00 | 100.00 | 94.91 | 87.55 | 60.59 | 76.55 |
| **CaliDrop_Full** | 100.00 | 100.00 | 94.18 | 86.27 | 61.91 | **76.80** |

---

> ### Author Response · Authors · 2025-11-24
>
> # General Response: Efficiency, Optimization, and Long-Context Scalability (Part Ⅱ)
>
>
> ### 4. Efficiency Evaluation
>
> To validate the effectiveness of the optimized CaliDrop, we conduct a comprehensive efficiency evaluation. While Figure 3(a) in the main paper presents results with varying batch sizes under a fixed input length, we here provide the results of experiments fixing the batch size and scaling the input length. Using LLaMA-3.1-8B-Instruct on an NVIDIA A800 GPU, we fix the batch size at 4 and the output length at 256. We compare the efficiency of our method (`calibration_size`=4096) against FullKV and SnapKV, recording Time-to-First-Token (TTFT), total latency, throughput, peak GPU memory, and peak CPU memory usage.
>
> **Key Observations:**
> 1.  **Memory:** CaliDrop incurs modest CPU memory overhead (utilizing abundant RAM) while keeping GPU memory usage comparable to SnapKV and significantly lower than FullKV (which OOMs at 32k).
> 2.  **Latency/Throughput:** While CaliDrop has a slight overhead compared to SnapKV due to calibration, it maintains a massive advantage over FullKV. For example, at 16k context, CaliDrop is **~2.5x faster** than FullKV in throughput (58.38 vs 23.24 tok/s).
>
> These results demonstrate that with proper optimization (Partial Offloading + Int8), CaliDrop provides a highly favorable trade-off: significant accuracy recovery for long contexts with manageable efficiency costs.
>
> **Table 3: Efficiency Metrics (Batch Size = 4, Output Length = 256)**
>
> | Metric | Method | 1024 | 4096 | 8192 | 16384 | 32768 |
> | :--- | :--- | :---: | :---: | :---: | :---: | :---: |
> | **TTFT (ms)** | FullKV | 333.81 | 1390.52 | 2981.04 | 6669.00 | *OOM* |
> | | SnapKV | 367.17 | 1488.15 | 3086.43 | 6771.54 | 16400 |
> | | **Ours** | 389.71 | 1576.28 | 3218.90 | 6927.72 | 16721 |
> | **Throughput (tok/s)** | FullKV | 135.63 | 72.50 | 42.85 | 23.24 | *OOM* |
> | | SnapKV | 131.10 | 120.97 | 104.35 | 75.01 | 43.94 |
> | | **Ours** | 116.83 | 83.50 | 71.81 | 58.38 | 37.64 |
> | **Peak GPU Mem (GB)** | FullKV | 17.51 | 24.79 | 34.61 | 54.25 | *OOM* |
> | | SnapKV | 15.98 | 17.29 | 19.11 | 22.75 | 30.03 |
> | | **Ours** | 16.05 | 17.80 | 19.96 | 23.06 | 30.65 |
> | **Peak CPU Mem (GB)** | FullKV | 1.35 | 1.33 | 1.36 | 1.34 | *OOM* |
> | | SnapKV | 1.45 | 1.46 | 1.43 | 1.43 | 1.45 |
> | | **Ours** | 3.04 | 5.29 | 5.50 | 5.50 | 5.49 |
>
> ### Conclusion
> In summary, the **Optimized CaliDrop** effectively resolves previous efficiency concerns. By establishing a flexible trade-off between CPU memory capacity and GPU computational speed, our method transforms the "memory wall" bottleneck into a manageable and amortized latency cost. We believe this makes CaliDrop a practical, scalable, and highly effective solution for deploying long-context LLMs on resource-constrained hardware.

---

### Author Response · Authors · 2025-11-29

# To Area Chair: Summary of Rebuttal Updates

**Dear Area Chair,**

We understand the challenges caused by the recent revert. To assist your assessment, we summarize below the interactions during the discussion period. Notably, we provided an **Optimized CaliDrop** implementation (Partial Offloading + Int8 Quantization + Fused Kernels) which successfully resolved the primary efficiency concerns shared by three reviewers, leading to an **explicit score raise from Reviewer nBZu** and **positive feedback from Reviewer dEEK**.

### Reviewer nBZu
* **Score:** Initial **4**, Final **6**
* **Initial Review:** Praised the method as "straightforward and sensible" with "noticeable benefits." The main concerns were practical efficiency (latency/offloading overhead) and a lack of ultra-long context experiments.
* **Author Response:** We introduced **Optimized CaliDrop** (incorporating Partial Offloading and Int8 Quantization) and provided new benchmarks showing **~2.5x higher throughput** than FullKV and success on **128k context lengths**.
* **Reviewer Follow-up (Update score from 4 to 6):**  The reviewer actively replied, stating they were "impressed by the ability of the authors to provide a viable solution to the latency issue" and updated their score to reflect the improved efficiency.
* **Author Final:** Promised to add the revision to the camera ready version and addressed the minor questions regarding quantization trade-offs.

### Reviewer ygNM
* **Score:** Initial **8**, Final **8**
* **Initial Review:** Rated the work as "**Excellent**" and well-motivated. Concerns focused on hyperparameter sensitivity ($\theta_1, \theta_2$) and potential memory peaks during recomputation.
* **Author Response:** We demonstrated empirical stability across parameter ranges and introduced the `calibration_size` hyperparameter for enhanced control. We also clarified that the layer-wise recomputation ensures no GPU memory peaks occur even with long contexts.

### Reviewer yoKp
* **Score:** Initial **4**, Final **4** (Concerns Addressed)
* **Initial Review:** Recognized the method as "simple yet effective." Concerns included uncertainty about applicability to the decoding phase, a lack of quantitative GPU memory overhead data, and layer-wise influence of CaliDrop.
* **Author Response:** We clarified that CaliDrop is **fully active during decoding** (streaming). We provided a detailed memory breakdown (Table 3 in General Response), proving that Optimized CaliDrop incurs **negligible overhead** compared to SnapKV. We add additional experiments about layer-wise influence of CaliDrop and found that the calibration in middle layers is more important.

### Reviewer dEEK
* **Score:** Initial **4**, Final **4** (Concerns Addressed & Positive Feedback)
* **Initial Review:** Appreciated the "novel insight." Initially concerned about the calibration/recomputation ablation and requested efficiency metrics.
* **Author Response:** Clarified the synergistic nature of calibration/recomputation and provided detailed efficiency metrics.
* **Reviewer Follow-up (Positive):** The reviewer acknowledged our response, agreeing that while offloading introduces overhead compared to pure eviction, **"the accuracy improvement is significant in small token budget."** They also suggested comparing with ShadowKV.
* **Author Final:** We clarified that CaliDrop (algorithmic importance identification) and ShadowKV (system-level optimization) are **complementary research lines**, and CaliDrop can effectively enhance such hardware-centric methods.


### Conclusion
The rebuttal period has strengthened our paper significantly. The **Optimized CaliDrop** has proven to be a robust solution for **128k contexts** with real-time efficiency. Given the **Strong Accept (8)** from Reviewer ygNM, the **explicit score raise (4 $\to$ 6)** from Reviewer nBZu, and the **positive feedback** from Reviewer dEEK upon validating these optimizations, we are confident the paper meets the high standards of ICLR.

Best regards,

**The Authors**

---

### Meta-Review · Area_Chair_wUa4 · 2025-12-17

**Summary:**

(*Disclaimer: given the peculiar review process, some of my choices and reasonings below will be highly subjective, as I tried to imagine how a reviewer would have reacted to a specific response. I understand that any negative choice will be perceived as unfair by the authors, and I apologize in advance for that.*)

(*Second disclaimer: the authors and some reviewers explicitly mention some changes in scores that occurred during the rebuttal. As these were reverted due to the possibility of collusion in light of the security incident, I will tend to disregard this information.*)

The paper proposes a novel method for improving KV cache selection mechanisms, based on a "calibration" phase that allows to merge an estimate of the attention scores belonging to the removed KV pairs. This calibration is done at fixed steps (using the previous evicted tokens), as the authors show that similar queries have similar attention scores.

The four initials reviews were clustered towards a weak rejection, will only one reviewer proposing a strong acceptance. The two major concerns were (a) limited evaluations, especially in the long-context case, and (b) the need for offloading KV pairs to the CPU RAM, which creates non-negligible memory issues.

Concerning point (b), the authors also acknowledge it to be an issue. In the rebuttal, they introduced significant modifications to the paper (most notably an optimised implementation of the method) and new results. As highlighted by `nBZu`, "*These modifications to the original method will probably lead to non-negligible updates in the camera-ready version*". The reviewer also had additional concerns based on these updates that were not answerable in the rebuttal period.

Of the other three reviewers, two did not interact during the rebuttal period, while one reviewer (`dEEK`) remained concerned, as detailed below. I personally agree with the view that the modifications are too large to be feasibly handled in a rebuttal, and would require a resubmission. Given that the reviewers (even in an optimistic scenario) would not have reached a consensus over acceptance, I vote for rejecting the current version of the manuscript.

**Reviewer Concerns:**

(*I will focus on some key weaknesses identified by multiple reviewers.*)

**Effect of CPU offloading** (`dEEK`, `yoKp`, `ygNM`, `nBZu`): this is the main concern, shared across all reviewers. As argued above, the authors introduced significant additions (also in the implementative side), but these raise additional questions and concerns that are not viable for a short rebuttal.

**Additional ablations and experiments** (e.g., long-context) (`dEEK`, `yoKp`, `ygNM`, `nBZu`): different reviewers asked for difference comparisons and ablations. The authors made several additions that answer most of these concerns.

**Prefilling vs. decoding** (`yoKp`): currently, the method is only shown in the prefilling phase. The authors mention that "*CaliDrop can be extended to the decoding phase with specific modifications*", but they do not show results in this setting. While the reviewer did not interact further, I am not convinced that "*high cosine similarity between queries is an intrinsic property of the model*" as the authors argue without further evidence from the experiments.

**Reviewer Scores:**

`dEEK`: while the discussion was interrupted, the reviewer was unconvinced by several answers from the authors. I do not believe their score would have been raised.

`yoKp`: they did not answer in the rebuttal. They might or might not have increased their score to a weak acceptance.

`ygNM`: the only reviewer immediately voting for a strong acceptance. It should be noted this was a relatively shallow review, as also evidenced by several concerns raised by the other reviewers.

`nBZu`: while the reviewer was satisfied by the improvements, they also acknowledged the potential need for additional reviews given the significant modifications to the paper.

---

### Decision · Program_Chairs · 2026-01-26

Reject